# Scalable and Equivariant Spherical CNNs by Discrete-Continuous (DISCO) Convolutions

**Jeremy Ocampo**[1,2]**, Matthew A. Price**[1,2]**, Jason D. McEwen**[1,2*]
[1]Kagenova Limited, [2]University College London (UCL)

## Abstract

No existing spherical convolutional neural network (CNN) framework is both computationally scalable and rotationally equivariant. Continuous approaches capture rotational equivariance but are often prohibitively computationally demanding. Discrete approaches offer more favorable computational performance but at the cost of equivariance. We develop a hybrid discrete-continuous (DISCO) group convolution that is simultaneously equivariant and computationally scalable to high-resolution. While our framework can be applied to any compact group, we specialize to the sphere. Our DISCO spherical convolutions exhibit $SO(3)$ rotational equivariance, where $SO(n)$ is the special orthogonal group representing rotations in $n$-dimensions. When restricting rotations of the convolution to the quotient space $SO(3)/SO(2)$ for further computational enhancements, we recover a form of asymptotic $SO(3)$ rotational equivariance. Through a sparse tensor implementation we achieve linear scaling in number of pixels on the sphere for both computational cost and memory usage. For 4k spherical images we realize a saving of $10^9$ in computational cost and $10^4$ in memory usage when compared to the most efficient alternative equivariant spherical convolution. We apply the DISCO spherical CNN framework to a number of benchmark dense-prediction problems on the sphere, such as semantic segmentation and depth estimation, on all of which we achieve the state-of-the-art performance.

## 1 Introduction

Spherical data are prevalent across many fields, from the relic radiation of the Big Bang in cosmology, to $360°$ imagery in virtual reality and computer vision. High-resolution data on the sphere are increasingly common in these and many other fields. Existing deep learning techniques for spherical data, however, cannot scale to high-resolution while also exhibiting equivariance, a powerful inductive bias responsible in part for the success of Euclidean convolutional neural networks (CNNs). Furthermore, many tasks involve dense predictions (e.g. semantic segmentation, depth estimation), necessitating pixel-wise outputs from deep learning models, exacerbating computational challenges.

**Continuous spherical CNNs approaches.** Bronstein et al. (2021) present the categorization of geometric deep learning approaches. Deep learning on the sphere falls into the *group* category, since the sphere is a homogeneous space with global symmetries on which the group of 3D rotations $SO(3)$ acts. In fact, the sphere is often considered as the prototypical example of the group setting. A number of group-based spherical CNN constructions have been developed (Cohen et al., 2018; Kondor et al., 2018; Esteves et al., 2018; 2020; Cobb et al., 2021; McEwen et al., 2022; Mitchel et al., 2022), where Fourier representations of spherical signals (i.e. spherical harmonic representations), combined with sampling theorems on the sphere (Driscoll & Healy, 1994; McEwen & Wiaux, 2011), are considered to provide access to the underlying continuous spherical signals and symmetries. While such approaches live natively on the sphere and capture rotational equivariance, they are highly computationally costly due to the need to compute spherical harmonic transforms (while fast spherical harmonic transforms exist they remain computationally demanding). McEwen et al. (2022) develop scattering networks on the sphere to alleviate computational considerations. While such an approach helps to scale to high-resolution input data, some high-resolution information is inevitably lost and architectures providing dense predictions are not supported.

---

*Corresponding author: `jason.mcewen@kagenova.com`

**Discrete spherical CNNs approaches.** Other approaches to spherical CNNs generally fall under the *grid*, *graph* or *geodesic* geometric deep learning categories yielding discrete approaches (Boomsma & Frellsen, 2017; Jiang et al., 2019; Zhang et al., 2019; Perraudin et al., 2019; Cohen et al., 2019). These approaches offer more favorable computational performance than the group-based frameworks but at the cost of equivariance. Since a completely regular point distribution on the sphere does in general not exist, these discrete approaches lose the connection to the underlying continuous symmetries of the sphere and thus cannot fully capture rotational equivariance (although in some cases limited rotational equivariance may be achieved; e.g. Cohen et al. 2019).

**Our approach.** Previous spherical CNN constructions can be categorised into two broad approaches: *continuous approaches* that capture rotational equivariance but are computationally demanding; and *discrete approaches* that do not fully capture rotationally equivariance but offer improved computational scaling. In this article we develop a novel *hybrid discrete-continuous (DISCO) approach* that is simultaneously computationally scalable to high-resolution, while also exhibiting excellent equivariance properties (see Figure 1). We define a DISCO group convolution, which we then specialize to the sphere. A transposed DISCO convolution can be considered to support dense-prediction tasks; hence, both high-resolution inputs and outputs are supported by our framework. DISCO convolutions afford a computationally scalable implementation through sparse tensor representations. We build DISCO spherical CNNs that achieve state-of-the-art performance on a number of high-resolution dense prediction tasks, including semantic segmentation and depth estimation.

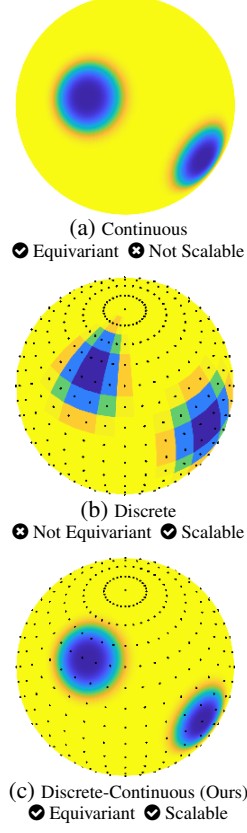

(a) Continuous
✅ Equivariant ❌ Not Scalable

(b) Discrete
❌ Not Equivariant ✅ Scalable

(c) Discrete-Continuous (Ours)
✅ Equivariant ✅ Scalable

Figure 1: Spherical CNN categorization.

## 2 BACKGROUND

### 2.1 CONTINUOUS GROUP CONVOLUTION

**Definition.** To generalize CNNs to group geometric deep learning settings the group convolution can be considered (see, e.g., Cohen & Welling, 2016; Esteves, 2020) where the convolution (sometimes called correlation) of two functions $f, \psi : G \to \mathbb{R}$ on the group $G$ is given by

$$(f \star \psi)(g) = \int_{u \in G} f(u)\psi(g^{-1}u)\mathrm{d}\mu(u), \tag{1}$$

where $g, u \in G$ and $\mathrm{d}\mu(u)$ is the invariant Haar measure. In many cases signals are not defined on a space with a group structure. Often signals are defined on a quotient space[1] $G/H$, where $H$ is a subgroup of $G$, i.e. $u \in G/H$ and $f, \psi : G/H \to \mathbb{R}$. For Lie groups, if $H$ is non-normal then $G/H$ is not a group but simply a differentiable manifold on which $G$ acts, i.e. a homogeneous space. In either setting the group convolution exhibits equivariance to group actions, that is $(\mathcal{Q}f \star \psi)(g) = (\mathcal{Q}(f \star \psi))(g)$, where $\mathcal{Q}$ denotes the group action corresponding to $q \in G$, i.e. $(\mathcal{Q}f)(g) = f(q^{-1}g)$.

**Fourier representation and sampling theory.** The group convolution may be written in Fourier space by the product $\widehat{(f \star \psi)}(k) = \widehat{f}(k)\widehat{\psi}^*(k)$ (see, e.g., Esteves, 2020), where $k$ is the Fourier conjugate variable of $u$, $\widehat{\cdot}$ denotes the Fourier transform, and $\cdot^*$ denotes complex conjugation. For compact manifolds (groups) the Fourier space is discrete (the canonical example being the Fourier series of functions defined on the circle $S^1$, i.e. periodic functions). Furthermore, for bandlimited signals on compact homogeneous manifolds, exact quadrature formulae exist (Pesenson & Geller, 2011), from which the existence of a sampling theorem follows where all information content of a continuous bandlimited signal can be captured in a finite number of spatial samples.

**Computation.** These results provide a strategy to compute the continuous group convolution *exactly* for bandlimited functions defined on compact homogeneous manifolds: (i) compute the finite

---

[1]The quotient space $G/H$ can be considered (in a non-rigorous way) as formed by collapsing all elements of $H$ to the identity in $G$, in a sense factoring out the elements of $H$ from $G$.

set of Fourier coefficients of the signal and filter leveraging the sampling theorem; (ii) compute the group convolution as a product in Fourier space; (iii) compute the convolved signal from its Fourier coefficients. Although the computation is performed in a discrete manner through the finite Fourier representation, the underlying continuous group convolution is computed exactly, without any discretization error. Consequently, perfect group equivariance is achieved. Note, however, that this approach does require computation of the Fourier transform on the manifold, which may be costly.

## 2.2 Continuous Spherical Convolutions

The strategy outlined above to compute the continuous group convolution exactly for bandlimited signals on compact homogeneous manifolds is the generalization of precisely the approach taken for group-based spherical CNNs (Cohen et al., 2018; Kondor et al., 2018; Esteves et al., 2018; 2020; Cobb et al., 2021; McEwen et al., 2022), where Fourier (i.e. spherical harmonic) representations of signals are computed to provide access to the underlying continuous structure and symmetries of the sphere. The corresponding spherical convolutions exhibit perfect rotational equivariance, with any numerical errors arising simply from finite precision arithemetic (see, e.g., Cobb et al., 2021). We concisely review the details of this approach to constructing spherical CNNs.

**Definition.** The spherical convolution of two functions $f, \psi : S^2 \to \mathbb{R}$ is given by

$$(f \star \psi)(R) = \int_{S^2} f(\omega)\psi(R^{-1}\omega)\mathrm{d}\mu(\omega), \tag{2}$$

where $\omega = (\theta, \phi)$ denote spherical coordinates with colatitude $\theta \in [0, \pi]$ and longitude $\varphi \in [0, 2\pi)$, and $\mathrm{d}\mu(\omega) = \sin\theta\mathrm{d}\theta\mathrm{d}\phi$ is the Haar measure. We consider rotations $R \in \mathrm{SO}(3)$, where $\mathrm{SO}(n)$ is the special orthogonal group of rotations in $n$-dimensions. The sphere $S^2$ is isomorphic to the quotient space $\mathrm{SO}(3)/\mathrm{SO}(2)$; it does not exhibit a group structure but is a homogeneous manifold.

**Fourier representation and sampling theory.** Since both $S^2$ and $\mathrm{SO}(3)$ are compact, their Fourier spaces are discrete and sampling theorems can be leveraged to compute Fourier representations exactly for sampled bandlimited signals (Driscoll & Healy, 1994; Kostelec & Rockmore, 2008; McEwen & Wiaux, 2011; McEwen et al., 2015). The spherical convolution can then be expressed as a product in Fourier space (see, e.g., McEwen et al., 2013; Cohen et al., 2018; Cobb et al., 2021).

**Computation.** For signals on the sphere bandlimited at $L$ (i.e. with Fourier coefficients zero above $L$), spherical convolutions can be computed exactly as a product in Fourier space. Although the computation is performed in a discrete manner through a Fourier representation, by appealing to underlying sampling theorems the continuous convolution is computed exactly. Consequently, perfect rotational equivariance is achieved. However, the harmonic transforms required are computationally demanding (even with fast algorithms; e.g. McEwen & Wiaux 2011; McEwen et al. 2015) and present the major computational bottleneck in scaling equivariant spherical CNNs to high-resolution.

## 2.3 Discrete Spherical Convolutions

Alternative approaches to constructing spherical CNNs typically consider a discrete approximation of the convolution of Equation 2 (Boomsma & Frellsen, 2017; Jiang et al., 2019; Zhang et al., 2019; Cohen et al., 2019). It is well-known that a completely regular point distribution on the sphere does in general not exist. Consequently, while a variety of spherical discretization schemes exists, it is not possible to discretize the sphere in a manner that is invariant to rotations (Cobb et al., 2021). On the other hand, since discrete approaches avoid the need for Fourier transforms on $S^2$ and $\mathrm{SO}(3)$, they are generally more computationally efficient than the continuous approaches discussed above that do exhibit rotational equivariance.

## 3 Discrete-Continuous (DISCO) Convolutions

To simultaneously realize computationally scalability and rotational equivariance we require a hybrid convolution that avoids Fourier transforms on manifolds, which otherwise induce considerable computational cost, while also avoiding a complete discretization, which otherwise sacrifices rotational equivariance. We present a novel DISCO (discrete-continuous) group convolution that achieves precisely this aim. We then specialize to the spherical setting, before discussing equivariance, filter parameterization and the corresponding transposed convolution for dense predictions.

## 3.1 DISCO GROUP CONVOLUTION

The DISCO group convolution follows by a careful hybrid representation of the group convolution of Equation 1. Some components of the representation are left continuous, to facilitate accurate rotational equivariance, while other components are discretized, to yield scalable computation.

**Definition.** We carefully approximate the group convolution by the DISCO representation

$$(f \star \psi)(g) = \int_{u \in G} f(u)\psi(g^{-1}u)\mathrm{d}\mu(u) \approx \sum_i f[u_i]\psi(g^{-1}u_i)q(u_i), \tag{3}$$

where square brackets and index subscripts denote discretized quantities and round brackets denote continuous quantities. Clearly the signal of interest must be discretized at sample positions $u_i$, where $i$ denotes the sample index. Critically, however, the filter $\psi$ and the group action remain continuous. This allows the filter $\psi$ to be transformed continuously by any $g$, keeping a coherent representation that avoids any discretization errors and, consequently, affords group equivariance, unlike a fully discrete method. The integral with respect to $u$ is also discretized. For bandlimited signals on compact homogeneous manifolds, the existence of a sampling theorem ensures that the integral can be approximated accurately using quadrature weights $q(u_i)$ (see Section 2.1).

**Approximation accuracy and equivariance.** The DISCO approximation of Equation 3 is highly accurate for bandlimited signals (real-world signals can be well approximately by bandlimited signals for a sufficient bandlimit). By appealing to a sampling theorem, all information content of the signal can be captured in the finite set of samples $\{f[u_i]\}$. The filter is represented continuously so does not introduce any error. The only source of approximation error is thus the quadrature used to evaluate the integral. For a sufficiently dense sampling, one can appeal to the sampling theorem and corresponding quadrature to evaluate this exactly. Therefore, it is possible in principle to compute the DISCO group convolution exactly, without any approximation error. Since the approximation is highly accurate, which can be made exact for a sufficiently dense sampling, and group actions are treated continuously, the DISCO group convolution exhibits excellent equivariance properties.

**Computational considerations.** The DISCO convolution also affords a scalable implementation. Firstly, it avoids any need for costly Fourier transforms on manifolds and groups. Secondly, as in the Euclidean setting, filters with local spatial support are considered to reduce both memory and computational requirements. The naive scaling of the DISCO group convolution for a general kernel is $O(N^2)$, where for simplicity $N$ represents both the number of input and output samples. However, for localized filters with $K$ non-zero samples, where $K \ll N$, the scaling reduces to $O(KN) = O(N)$, just as for Euclidean CNNs.

## 3.2 DISCO SPHERICAL CONVOLUTION

**Definition.** We now specialize to the spherical setting, where the spherical convolution of Equation 2 can be accurately approximated by the DISCO representation

$$(f \star \psi)(R) = \int_{S^2} f(\omega)\psi(R^{-1}\omega)\mathrm{d}\mu(\omega) \approx \sum_i f[\omega_i]\psi(R^{-1}\omega_i)q(\omega_i), \tag{4}$$

where, for now, we consider 3D rotations $R \in \mathrm{SO}(3)$, i.e. $f \star \psi : \mathrm{SO}(3) \to \mathbb{R}$. As in the general group setting the signal must clearly be discretized at sample positions $\omega_i$, as is the integral with quadrature weights $q(\omega_i)$. Again, critically, the filter $\psi$ and group action, in this case corresponding to rotation $R$, remain continuous, facilitating rotational equivariance.

**Approximation accuracy and equivariance.** We appeal to the sampling theorem on the sphere of McEwen & Wiaux (2011) since it provides the most efficient sampled signal representation (halving the Nyquist rate compared to Driscoll & Healy 1994). For spherical signals bandlimited at $L$, all information content of the underlying continuous signal is captured in $\sim 2L^2$ samples $\{f[\omega_i]\}$. The only source of approximation error is then the discretization of the integral. By adopting exact quadrature (McEwen & Wiaux, 2011) this integral could in principle be computed exactly (see Appendix A). However, since the integrand is the product of two signals each bandlimited at $L$, it is itself bandlimited at $2L$. Exact quadrature would thus require the signals to be sampled at $\sim 2(2L)^2 = 8L^2$ positions. For our implementation we avoid upsampling signals, trading off a small approximation error to avoid a four-fold increase in computation. Nevertheless, we do adopt

the quadrature weights of McEwen & Wiaux (2011) to keep approximation errors to a minimum. Since the approximation remains highly accurate, and the action of rotations on filters is treated continuously, the DISCO spherical convolution exhibits accurate SO(3) equivariance (see Section 3.3).

**Computational considerations.** For the continuous spherical convolution approach computed via Fourier space, the computation cost is dominated by the Fourier transform on the sphere, which naively is $O(N^2)$ but for fast algorithms scales as $O(N^{3/2})$ (e.g. McEwen & Wiaux, 2011). The naive computational scaling of the DISCO spherical convolution is $O(N^2)$. When considering localized filters with on average $K$ non-zero terms, with $K \ll N$, the scaling of the DISCO spherical convolution is reduced to $O(N)$, again just as in the Euclidean setting.

**Restricting rotations to SO(3)/SO(2).** While the DISCO spherical convolution is already efficient, we seek further computational savings by reducing the space of rotations considered from SO(3) to SO(3)/SO(2) (we provide a visual illustration of SO(3) and SO(3)/SO(2) rotations in Fig. 3). To make this explicit, first consider the parameterization of a rotation $R \in$ SO(3) by the Euler angles $\alpha \in [0, 2\pi)$, $\beta \in [0, \pi]$ and $\gamma \in [0, 2\pi)$, where the rotation may be written $R = Z(\alpha)Y(\beta)Z(\gamma)$, following the $zyz$ Euler convention for rotations $Z(\cdot)$ and $Y(\cdot)$ about the $z$ and $y$ axes, respectively. Rotations restricted to the quotient space may be written $R = Z(\alpha)Y(\beta) \in$ SO(3)/SO(2). Restricting rotations to SO(3)/SO(2), which is isomorphic to the sphere $S^2$, is analogous to typical Euclidean planar CNNs, where filters are translated across the image but are *not* rotated in the plane. However, as the space SO(3)/SO(2) is not a group, when restricting rotations in this manner important differences to the usual setting arise since we no longer have a group convolution (as elaborated in Section 3.3). We denote the spherical convolution when restricting to $R \in$ SO(3)/SO(2) by $f \circledast \psi :$ SO(3)/SO(2) $\to \mathbb{R}$, using a different symbol to the SO(3) spherical convolution to avoid any confusion. Since SO(3) is a three dimensional space, whereas SO(3)/SO(2) is two dimensional, restricting rotations to SO(3)/SO(2) reduces the dimension of the output space from $O(L^3)$ to $O(L^2)$, yielding an overall computational scaling of $O(N) = O(L^2)$, where we recall $L$ denotes the spherical signal bandlimit.

## 3.3 ROTATIONAL EQUIVARIANCE

**SO(3) rotational equivariance.** Consider the DISCO spherical convolution $f \star \psi$ for rotation $R \in$ SO(3), with $\mathcal{Q}$ denoting the action of the rotation $Q \in$ SO(3) on signals, which we show satisfies SO(3) rotational equivariance:

$$((\mathcal{Q}f) \star \psi)(R) \approx \sum_i (\mathcal{Q}f)[\omega_i]\psi(R^{-1}\omega_i)q(\omega_i) \overset{(*)}{=} \sum_i f[\omega_i]\psi((Q^{-1}R)^{-1}\omega_i)q(\omega_i)$$

$$\overset{(**)}{\approx} (f \star \psi)(Q^{-1}R) = (\mathcal{Q}(f \star \psi))(R). \quad (5)$$

The only approximations involved in Equation 5 (highlighted by the approximate equality symbols) follow from the quadrature of the integral. Critically, no approximation is involved in applying the rotation $Q$ due to the continuous representation of rotations and the filter $\psi$, i.e. the equality labeled $(*)$ is exact. Consequently, the DISCO spherical convolution $f \star \psi$ exhibits excellent SO(3) rotational equivariance. An important point to note in the SO(3) rotational equivariance proof, which is often ignored, is that it is only possible to relate the expression back to a spherical convolution, i.e. the step labeled $(**)$, since SO(3) exhibits a group structure and so $Q^{-1}R \in$ SO(3).

**Asymptotic SO(3) rotational equivariance.** Consider the DISCO spherical convolution $f \circledast \psi$ and rotations $R \in$ SO(3)/SO(2). Since SO(3)/SO(2) is not a group, $Q^{-1}R \notin$ SO(3)/SO(2) and step $(**)$ of the equivariance proof of Equation 5 does not extend to $f \circledast \psi$. Consequently, $f \circledast \psi$ for $R \in$ SO(3)/SO(2) does in general not satisfy SO(3) or SO(3)/SO(2) equivariance (contrast with the Euclidean planar setting that is translationally equivariant for the analogous setting). However, the DISCO spherical convolution $f \circledast \psi$ does satisfy a form of asymptotic SO(3) equivariance. Let $Q = Z(\alpha)Y(\beta)Z(\gamma)$ and $R = Z(\alpha')Y(\beta')$, then $Q^{-1}R = Z(-\gamma)Y(-\beta)Z(\alpha'-\alpha)Y(\beta') \in$ SO(3). As $\beta \to 0$, we have $Q^{-1}R \to Z(\alpha'-\alpha-\gamma)Y(\beta') \in$ SO(3)/SO(2) and hence $((\mathcal{Q}f) \circledast \psi)(R) \to (\mathcal{Q}(f \circledast \psi))(R)$, i.e. we recover asymptotic SO(3) rotational equivariance as $\beta \to 0$. Note that there is no restriction on $\alpha$ or $\gamma$, i.e. we acheive full rotational equivariance for rotations about the $z$ axis. This asymptotic SO(3) equivariance is validated numerically in Section 5.1 and shown to be reasonably accurate even for moderately large $\beta$. Asymptotic SO(3) equivariance is of significant practical use since content in spherical signals is often orientated and similar content often appears

at similar latitudes, particularly for $360°$ panoramic photos and video. Hence, asymptotic $SO(3)$ rotational equivariance may even be more desirable than $SO(3)$ equivariance for certain applications.

**Axisymmetric filters and SO(3) rotational equivariance.** The rotational equivariance considerations presented above hold for general filters with directional structure. Axisymmetric filters, i.e. filters that are invariant to rotations about their own axis, may also be considered. While axisymmetric filters are less expressive than directional filters they have nevertheless been shown to be effective for spherical CNNs (Esteves et al., 2018). For axisymmetric filters the $SO(3)$ spherical convolution $f \star \psi$ and the $SO(3)/SO(2)$ spherical convolution $f \circledast \psi$ identify since $f \star \psi$ for $R = Z(\alpha)Y(\beta)Z(\gamma) \in SO(3)$ is independent of $\gamma$. Consequently, the DISCO spherical convolution $f \circledast \psi$ for $R \in SO(3)/SO(2)$ with an axisymmetric filter $\psi$ does exhibit $SO(3)$ rotational equivariance (as validated numerically in Section 5.1).

### 3.4 FILTER PARAMETERIZATION

We consider a continuous filter representation $\psi$ in the DISCO spherical convolution, which allows us to evaluate $\psi(\omega)$ for any coordinate $\omega \in S^2$ and thus compute $\psi(R^{-1}\omega)$ for any continuous rotation $R$ exactly. Nevertheless, the filter must be parameterized by a finite number of learnable parameters $\boldsymbol{p}$. We consider filters that are either axisymmetric $\psi(\theta; \boldsymbol{p})$, directional and separable $\psi(\theta, \phi; \boldsymbol{p}) = \vartheta(\theta; \boldsymbol{p})\varphi(\phi; \boldsymbol{p})$, or directional $\psi(\theta, \phi; \boldsymbol{p})$ without further constraints. In all cases the filters are parameterized with equally spaced nodes along $\theta$ and $\phi$, with each node value a learnable parameter. To evaluate the filter for any continuous argument we simply use linear interpolation. We localize the filter spatially by constraining it to be zero for angles above some cutoff $\theta_{\text{cutoff}}$.

### 3.5 DISCO TRANSPOSED SPHERICAL CONVOLUTION

Dense pixel-wise predictions are required for numerous deep learning tasks. Existing equivariant spherical CNNs cannot support dense predictions since they are too computationally demanding. A DISCO transposed spherical convolution can be defined to increase the resolution of internal feature representations in an analogous manner to the transposed convolutions considered in Euclidean U-Net architectures (Ronneberger et al., 2015), supporting dense predictions. The DISCO transposed spherical convolution $f \circledast^\dagger \psi : S^2 \to \mathbb{R}$, restricting rotations to $SO(3)/SO(2)$, is given by

$$(f \circledast^\dagger \psi)(\omega) = \int_{S^2} f(\omega')\psi(R_{\omega'}^{-1}\omega)\mathrm{d}\mu(\omega') \approx \sum_i f[\omega_i]\psi(R_{\omega_i}^{-1}\omega)q(\omega_i), \tag{6}$$

where we introduce the notation $R_\omega = Z(\phi)Y(\theta) \in SO(3)/SO(2)$ for $\omega = (\theta, \phi)$ in order to make the dependence of the rotation on the integration variable explicit. The output of the transposed convolution is the result of summing all filters rotated to each $R_{\omega_i}$, with a weight of $f[\omega_i]$, analogous to the Euclidean transposed convolution. Identical approximation and equivariance properties as the (non-transposed) DISCO spherical convolution hold.

## 4 COMPUTATIONALLY SCALABLE DISCO SPHERICAL CONVOLUTION

The DISCO convolution affords a computationally scalable implementation through sparse tensor representations. While we focus on the spherical setting of immediate practical interest, the same principles can be applied to any compact group. We leverage sparse-dense tensor multiplication operators to compute the DISCO spherical convolution efficiently on hardware accelerators (e.g. GPUs, TPUs). By exploiting symmetries of the sampling scheme we also describe how memory usage can be optimized. Automatic differentiation of sparse tensor operations that include learnable parameters is not supported in either TensorFlow or PyTorch, hence we derive custom sparse gradient representations to be used when training models.

### 4.1 SPARSE TENSOR REPRESENTATION

**Representation and sampling.** The DISCO spherical convolution for rotations $R \in SO(3)/SO(2)$, here denoted by the variable $h = f \circledast \psi$ for brevity, may be written by the tensor multiplication

$$h_j = \sum_i \Psi_{ji} f_i, \tag{7}$$

where $\Psi$ is a tensor with elements $\Psi_{ji} = \psi(R_j^{-1}\omega_i)$, $f_i$ is the sampled spherical input signal (collapsed to a vector) with quadrature weights absorbed, and $h_j$ is the convolved output at $R_j = Z(\phi_j)Y(\theta_j)$, or more simply, output coordinate $(\theta_j, \phi_j)$. We adopt the sampling theorem of McEwen & Wiaux (2011) so that all of the information content of bandlimited spherical signals can be captured in a finite set of $\sim 2L^2$ samples. We also adopt the associated quadrature weights but, as discussed in Section 3.2, do *not* upsample signals, trading off a small approximation error in numerical integration to realize a significant reduction in computational cost. While the numerical integration is then no longer exact, it remains highly accurate.

**Sparse representation.** To compute the elements of $\Psi$ we first compute the continuously rotated coordinates, which we denote as tensors $(\Theta_{ji}, \Phi_{ji}) = R_j^{-1}\omega_i$, and then pass them to the continuous filter function to evaluate $\Psi_{ji} = \psi(\Theta_{ji}, \Phi_{ji})$ without any discretization error. For spatially localized filters, $\Psi$ will be sparse and we need only compute the rotated coordinates if $\Psi_{ji}$ will be non-zero. We exploit knowledge of the support of the filters, given by $\theta_{\text{cutoff}}$, to limit the number of rotated coordinates that need to be computed. Consequently, $\Theta$ and $\Phi$ are also sparse tensors since we compute only coordinates where $\Psi$ will be non-zero. The sparse representation of $\Psi$ can thus be computed efficiently and Equation 7 can then be computed by a sparse-dense tensor multiplication. For transposed spherical convolutions, $\Psi$ is simply replaced by $\Psi^\dagger$. DISCO spherical convolutions and transpose convolutions can thus be computed efficiently by sparse-dense tensor operations that map well onto hardware accelerators, which for practical purposes can result in substantial enhancements in computational performance.

**Scalable computation.** Empirically we find that the computational cost of the DISCO spherical convolution scales linearly in the number of spherical pixels, i.e. as $O(N) = O(L^2)$, as predicted; furthermore, not only does it scale more favorably than the most efficient alternative equivariant spherical convolution layer but its absolute cost is also considerably lower (see Appendix B for details). For example, for 4k spherical images we achieve a saving in computational cost of $10^9$.

**Architectures.** Efficient spherical implementations of common CNN architectures can then be constructed by combining the DISCO forward and transpose spherical convolutions with pointwise non-linear activations and other common architectural features, such as skip connections, batch-normalization[2], multiple channels, etc. Note that in the architectures we consider we typically adopt depthwise separable convolutions (Chollet, 2017) for computational efficiency.

## 4.2 MEMORY OPTIMIZATIONS

**Explicit sampling.** For structured spherical sampling schemes many of the elements of $\Psi$ are repeated, which opens the possibility of memory compression. While memory may be compressed for various structured samplings, we focus on the sampling scheme that we adopt. Specifically, we sample spherical coordinates by $(\theta_t, \phi_p) = (\pi t/L, \pi p/L)$ for $t, p \in \mathbb{Z}$, $0 \le t \le L$, $0 \le p \le 2L-1$ (McEwen & Wiaux, 2011; Daducci et al., 2011), where recall $L$ is the spherical harmonic bandlimit, which plays the role of resolution, yielding $2L(L+1)$ pixels on the sphere. Incidentally, this sampling scheme maps onto the typical sampling of equirectangular 360° panoramic images well.

**Compressed tensor.** For clarity we write the input and output coordinates in terms of their individual spherical coordinates $\omega_i = (\theta_t, \phi_p)$ and $\omega_j = (\theta_{t'}, \phi_{p'})$, respectively, i.e. $i = (t, p)$ and $j = (t', p')$. It is then clear that the filter tensor $\Psi$ is a 4-dimensional tensor that may be written explicitly as $\Psi_{ji} = \Psi_{t'p'tp}$. By exploiting symmetries of our sampling scheme $\Psi$ can be reduced to 3 dimensions by noticing that the two tensor elements $\Psi_{t'p'tp}$ and $\Psi_{t'0t(p-p')}$ are equal. This follows since a rotation in $Z(\phi)$ is equivalent to a translation in the input coordinates by $\phi$:

$$\Psi_{t'p'tp} = \psi((Z(\phi_{p'})Y(\theta_{t'}))^{-1}(\theta_t, \phi_p)) = \psi(Y^{-1}(\theta_{t'})(\theta_t, \phi_p - \phi_{p'}')) = \Psi_{t'0t(p-p')}. \tag{8}$$

For our sampling scheme the set of all translated coordinates falls on the set of original coordinates, i.e. $\{p - p'\} = \{p\}$, and hence we can store $\Psi_{t'0t(p-p')}$ as $\Psi_{t'0tp}$. For a dense representation $\Psi_{t'0tp}$ has $O(L^3)$ elements; however, for localized filters this is reduced to $O(L^2)$ non-zero elements; hence, memory usage also scales linearly with number of spherical pixels.

---

[2]In order to ensure batch-normalization for sampled spherical signals is rotationally equivariant we adapt the standard batch-normalization to compute averages over the sphere, with appropriate sample weighting.

**Efficient compressed sparse-dense tensor multiplication.** The compressed version of $\Psi$ can be used efficiently in the sparse tensor multiplication by re-writing Equation 7 as

$$h_{t'p'} = \sum_{tp} \Psi_{t'p'tp} f_{tp} = \sum_{tp} \Psi_{t'0t(p-p')} f_{tp} = \sum_{tp} \Psi_{t'0tp} f_{t(p+p')}. \tag{9}$$

Notice that the last line is a multiplication between $f_{tp}$ shifted by $p'$ in the $p$ coordinate and $\Psi_{t'0tp}$. Hence, we can implement the compressed tensor multiplication by looping over $p'$ and simply shifting the input in $p$, which is considerably more efficient than repeatedly shifting the elements of a sparse tensor. To support multiple channels we adopt depthwise separable convolutions (Chollet, 2017) not only for computational efficiency, but also for memory efficiency.

**Scalable memory usage.** Empirically we find that when incorporating the optimizations discussed above memory requirements of the DISCO spherical convolution indeed scale linearly with number of spherical pixels, i.e. as $O(L^2)$, as predicted, providing a considerable saving over the most efficient alternative equivariant spherical convolution layer (see Appendix B for details). For example, for 4k spherical images we achieve a saving in memory of $10^4$.

### 4.3 SPARSE GRADIENTS

Deep learning models are typically trained by optimization algorithms that use gradients, which are computed efficiently by automatic differentiation. However, automatic differentiation of sparse tensor operations that include learnable parameters, as in the DISCO convolution, is not supported either by TensorFlow or PyTorch. Thus, we implement custom sparse gradients (see Appendix C).

## 5 EXPERIMENTS

We first demonstrate the equivariance properties of our scalable DISCO spherical convolutional layers, before tackling a number of dense-prediction benchmark problems with spherical CNNs built using our DISCO framework (implemented in the `CopernicAI` [3] code). We achieve state-of-the-art performance on all benchmarks. See Appendix E for further details on benchmark experiments.

### 5.1 EQUIVARIANCE TESTS

To test the equivariance of DISCO spherical convolutions we consider random test signals and compute the mean relative equivariance error between rotating signals before and after the convolution. For axisymmetric filters we achieve excellent rotational equivariance with error $\sim 0.04\%$. For directional filters, for which equivariance is asymptotic since we consider $\mathrm{SO}(3)/\mathrm{SO}(2)$ rotations, we recover equivariance errors of $\sim 0.01\%$ in the limit of latitudinal rotation $\beta = 0°$. As $\beta$ increases, equivariance errors increase but, nevertheless, we observe only moderate equivariance error of just a few percent at $\beta = 10°$ (this compares favorably to the equivariance error of $\sim 35\%$ for a spherical ReLU at $L = 32$; Cohen et al. 2018; Cobb et al. 2021). See Appendix D for further details.

### 5.2 ROTATED MNIST ON THE SPHERE

We consider the benchmark of classifying randomly rotated MNIST digits projected onto the sphere (Cohen et al., 2018) for three experimental modes NR/NR, R/R and NR/R, indicating whether the training/test sets, respectively, have been rotated (R) or not (NR). Results are presented in Table 1 for bandlimit $L = 1024$. The DISCO framework performs well at high resolution and, moreover, similar performance is achieved in all rotational settings, demonstrating excellent rotational invariance.

### 5.3 SEMANTIC SEGMENTATION

We consider the dense-prediction problem of semantic segmentation of $360°$ photos, using a common backbone of a DISCO spherical convolutional residual U-Net architecture, and compare to existing benchmark results (note that previous experiments have been performed at different resolutions; hence we quote the effective resolution $\tilde{L}$ determined by number of samples on the sphere).

---

[3] https://www.kagenova.com/products/copernicAI/

Table 1: Results for spherical MNIST.

|  | NR/NR | R/R | NR/R |
|---|---|---|---|
| Planar | 98.3 | 53.9 | 14.3 |
| **DISCO-Axisymmetric** | 98.6 | 98.7 | 98.6 |

Table 2: Results for 2D3DS.

| Model | mIoU | mAcc | $\tilde{L}$ |
|---|---|---|---|
| Planar UNet | 35.9 | 50.8 | 72 |
| UGSCNN (Jiang et al., 2019) | 38.3 | 54.7 | 72 |
| GaugeNet (Cohen et al., 2019) | 39.4 | 55.9 | 72 |
| HexRUNet (Zhang et al., 2019) | 43.3 | 58.6 | 72 |
| SWSCNNs (Esteves et al., 2020) | 43.4 | 58.7 | 91 |
| CubeNet (Shakerinava & Ravanbakhsh, 2021) | 45.0 | 62.5 | 83 |
| MöbiusConv (Mitchel et al., 2022) | 43.3 | 60.9 | 91 |
| TangentImg (Eder et al., 2020) | 41.8 | 54.9 | 286 |
| HoHoNet (Sun et al., 2021) | 43.3 | 53.9 | 256 |
| DISCO-Axisymmetric (Ours) | 39.7 | 54.1 | 256 |
| DISCO-Separable (Ours) | 43.9 | 60.9 | 256 |
| DISCO-Directional (Ours) | 45.2 | 61.5 | 256 |
| **DISCO-Directional-Aug (Ours)** | **45.7** | **62.7** | 256 |

Table 3: Results for Omni-SYNTHIA

| Model | mIoU | mAcc | $\tilde{L}$ |
|---|---|---|---|
| Planar UNet | 38.8 | 45.1 | 143 |
| UGSCNN (Jiang et al., 2019) | 36.9 | 50.7 | 143 |
| HexUNet (Zhang et al., 2019) | 43.6 | 52.2 | 143 |
| TangentImg (Eder et al., 2020) | 41.3 | 52.8 | 143 |
| Planar UNet | 44.6 | 52.6 | 286 |
| UGSCNN (Jiang et al., 2019) | 37.6 | 48.9 | 286 |
| HexUNet (Zhang et al., 2019) | 48.3 | 57.1 | 286 |
| TangentImg (Eder et al., 2020) | 35.8 | 55.3 | 286 |
| DISCO-Separable (Ours) | 48.3 | 59.3 | 256 |
| **DISCO-Separable-Aug (Ours)** | **49.2** | **63.7** | 256 |

Table 4: Results for Pano3D.

| Model | Parameters | Depth Error Metrics | | | | Depth Accuracy Metrics | | | |
|---|---|---|---|---|---|---|---|---|---|
|  |  | $w$RMSE | $w$RMSLE | $w$AbsRel | $w$SqRel | $\delta_{1.05}^{\text{ico}}$ | $\delta_{1.1}^{\text{ico}}$ | $\delta_{1.25}^{\text{ico}}$ | $\delta_{1.25^2}^{\text{ico}}$ |
| Planar UNet (Albanis et al., 2021) | 27M | **0.4520** | **0.1300** | 0.1147 | **0.0811** | 36.68% | 60.59% | 88.31% | 96.96% |
| **DISCO-Directional (Ours)** | **658k** | 0.5063 | 0.1695 | **0.1109** | 0.0852 | **38.32**% | **62.12**% | **88.65**% | **97.29**% |

**2D3DS dataset of indoor** 360° **photos.** Results for the semantic segmentation of indoor scenes of the 2D3DS dataset (Armeni et al., 2017) are presented in Table 2. Our DISCO-Directional architecture outperforms the DISCO-Axisymmetric architecture, which we attribute to being due to its asymptotic SO(3) equivariance (that can be considered as "orientation aware") and having more expressive directional filters. We also include augmentation, which helps to improve performance. We achieve state-of-the-art performance compared to all previous methods.

**Omni-SYNTHIA dataset of outdoor** 360° **photos.** Results for the semantic segmentation of outdoor city scenes of the Omni-SYNTHIA dataset (Ros et al., 2016) are presented in Table 3. We consider a DISCO-Separable architecture, with and without augmentation for ablation purposes. In both settings we achieve state-of-the-art performance.

## 5.4 DEPTH ESTIMATION

We consider the task of monocular depth estimation from 360° photos, tackling the Pano3D benchmark (Albanis et al., 2021) for the Matterport3D dataset (Chang et al., 2017). We adopt the same backbone architecture as for semantic segmentation. Results are presented in Table 4. Pano3D is a relatively new benchmark; the only previous results are those of Albanis et al. (2021). We achieve excellent performance with significantly fewer parameters ($40\times$ less). While we do not achieve the best performance across all metrics (which we suspect is due to the alternative model being trained with a virtual normal loss that leverages surface normal information), we perform better on accuracy metrics than error metrics and nevertheless achieve the state-of-the-art in six of nine metrics.

## 6 CONCLUSIONS

We have developed the DISCO (discrete-continuous) group convolution that is simultaneously computationally scalable and equivariant. While our framework can be applied to any compact group, we specialize to the sphere. We present a sparse tensor implementation with custom gradients that exhibits excellent rotational equivariance properties, is well-suited to hardware accelerators (e.g. GPUs, TPUs), and achieves linear scaling in both computational cost and memory usage. We apply our DISCO spherical CNN framework to numerous dense-prediction tasks, such as semantic segmentation and depth estimation, achieving state-of-the-art performance on all benchmarks.

ACKNOWLEDGMENTS

We thank Mayeul d'Avezac for general discussions regarding TensorFlow memory usage. We thank Chao Zhang for providing data processing code for the Omni-SYNTHIA benchmark. We thank Georgios Albanis and Nikolaos Zioulis for providing training details on the Pano3D benchmark.

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

## A   SAMPLING THEOREMS AND QUADRATURE ON THE SPHERE

Signals on the sphere $f \in L^2(S^2)$ may be decomposed into their harmonic representations by

$$f(\theta, \phi) = \sum_{\ell=0}^{\infty} \sum_{m=-\ell}^{\ell} f_{\ell m} Y_{\ell m}(\theta, \phi), \tag{10}$$

with spherical harmonic coefficients given by

$$f_{\ell m} = \int_{S^2} f(\theta, \phi) \, Y_{\ell m}^*(\theta, \phi) \, \mathrm{d}\omega(\theta, \phi), \tag{11}$$

where $Y_{\ell m}$ denote the spherical harmonic functions of natural order $\ell$ and integer degree $|m| \le \ell$. Consider signals on the sphere bandlimited at $L$, such that $f_{\ell m} = 0$ for all $\ell \ge L$. In this setting the summation over $\ell$ in Equation 10 may be truncated at $L - 1$:

$$f(\theta, \phi) = \sum_{\ell=0}^{L-1} \sum_{m=-\ell}^{\ell} f_{\ell m} Y_{\ell m}(\theta, \phi). \tag{12}$$

A sampling theorem on the sphere (e.g. Driscoll & Healy, 1994; McEwen & Wiaux, 2011) states how to sample a band-limited function $f(\theta, \phi)$ at a finite number of locations, such that all of the information content of the continuous function is captured. Since the sphere $S^2$ is a compact manifold, its Fourier space is discrete. All of the information content of a continuous bandlimited signal is thus captured in its finite set of harmonic coefficients $f_{\ell m}$ (for $0 \le \ell < L$ and $|m| \le \ell$). Indeed, once the finite set of $f_{\ell m}$ are known, the function can be evaluated for any continuous coordinate through Equation 12. A sampling theorem on the sphere is thus synonymous with the exact calculation of the spherical harmonic coefficients $f_{\ell m}$ of the continuous function from its samples. Consequently, sampling theorems effectively encode, often implicitly, an exact quadrature rule for evaluating the integral of a band-limited function on the sphere. In many cases it useful to have an explicit representation of the quadrature rule associated with a given sampling theorem.

In this work we adopt the sampling theorem on the sphere of McEwen & Wiaux (2011). Specifically, we adopt a variant of the sampling scheme first considered in Daducci et al. (2011) that oversamples by one in both $\theta$ and $\phi$ to recover samples that are diametrically opposite and thus exhibit antipodal symmetry (see Section 4.2 for explicit sample positions). This variant of the sampling scheme provides numerous practical advantages, such as mapping more closely onto the common equirectangular sampling of $360°$ photos and videos and supporting dyadic downsampling and upsampling.

While the explicit quadrature rule for the standard sampling scheme of McEwen & Wiaux (2011) is presented therein, the quadrature rule for the variant with antipodal symmetry that we adopt in this work has not appeared in any published literature; hence we present it here. Consider the following integral of a function $f(\theta, \phi)$ bandlimited at $L$:

$$I = \int_{S^2} f(\theta, \phi) \, \mathrm{d}\omega(\theta, \phi) = \sum_{t=0}^{L} \sum_{p=0}^{2L} f(\theta_t, \phi_p) q(\theta, \phi), \tag{13}$$

where $q(\theta, \phi)$ denote quadrature weights such that the integral $I$ is computed exactly from the sampled signal values $f(\theta_t, \phi_p)$. The derivation of the quadrature weights for the antipodally symmetric sampling scheme follows that given by (McEwen & Wiaux, 2011) for the standard sampling scheme, with only some minor variations.

The derivation of the sampling scheme of (McEwen & Wiaux, 2011) is based on an extension of the sphere to the torus through careful period extensions of the function of interest to the $\theta$ domain $[0, 2\pi)$; recall that on the sphere $\theta \in [0, \pi]$. Explicit quadrature weights can be determined by

folding back contributions from $(\pi, 2\pi)$ onto $(0, \pi)$. Consider the Fourier transform of the periodic function defined by $\sin\theta$ on $[0, \pi]$ and zero on $(\pi, 2\pi)$:

$$w(m') = \int_0^{\pi} \sin(\theta) \exp(im'\theta) \mathrm{d}\theta = \begin{cases} \pm i\pi/2, & m' = \pm 1 \\ 0, & m' \text{ odd}, m' \neq \pm 1 \\ 2/(1 - m'^2), & m' \text{ even} \end{cases} . \tag{14}$$

The bandlimited representation of this function is given by

$$w_{\mathrm{r}}(\theta_t) = \sum_{m'=-L}^{L-1} w(m) \exp(im'\theta_t) \tag{15}$$

(note the lower index of the summation of $-L$). The quadrature weights of the antipodally symmetric sampling scheme are then given by folding back contributions from $(\pi, 2\pi)$ onto $(0, \pi)$, with appropriate normalization:

$$q(\theta_t, \phi_t) = q(\theta_t) = \frac{2\pi}{(2L)^2} \big[ w_{\mathrm{r}}(\theta_t) + (1 - \delta_{t0})(1 - \delta_{tL})(-1)^s w_{\mathrm{r}}(\theta_{2L-t}) \big]. \tag{16}$$

For completeness we present the result above for signals with spin $s$, although in the current article we consider scalar signals with $s = 0$ only.

## B    COMPUTATIONAL COST AND MEMORY USAGE

We compute the theoretical computational cost and empirical memory usage of the DISCO spherical convolution, contrasting it to the most efficient alternative equivariant spherical convolutional layers of an axisymmetric convolution computed via Fourier space (Esteves et al., 2018; Cobb et al., 2021). Note that alternative equivariant spherical convolutional layers that are more expressive are considerably more costly, both in terms of compute and memory (Cohen et al., 2018; Kondor et al., 2018; Cobb et al., 2021).

Without loss of generality we consider 1 layer, 1 channel, and a batch size of 1. For the DISCO spherical convolutional layer we consider axisymmetric filters with 4 nodes and $\theta_{\mathrm{cutoff}} = 3\pi/L$. For the axisymmetric harmonic layer we parameterize the filters with 10 nodes in harmonic space and use linear interpolation, similar to Esteves et al. (2018).

For the DISCO layer we compute computational floating point (FLOP) cost via analytic calculations of the sparse tensor products and linear interpolation. For memory usage we explicitly construct the sparse $\Psi_{ji}$ tensor and then count the number of non-zero values; hence, the evaluation of memory usage is empirical rather than purely theoretical. For the axisymmetric harmonic layer we compute computational cost and memory usage via analytical calculations of the tensor products used in the precompute-based spherical harmonic transforms, convolutions and linear interpolation.

Theoretical computational cost and empirical memory usage results are presented in Fig. 2. For the DISCO spherical convolution both computational costs and memory usage scales linearly in the number of spherical pixels, i.e. as $O(N) = O(L^2)$. Furthermore, not only does the computational cost of the DISCO spherical convolution scale more favorably than the most efficient alternative equivariant spherical convolution layer but its absolute cost is also considerably lower. For 4k spherical images we achieve a saving in computational cost of $10^9$ and a saving in memory usage of $10^4$.

Inspecting the theoretical computational cost, as considered above, has the advantage that it abstracts away details of a particular implementation, its level of optimization, and the hardware on which the model runs. Wall-clock compute time, while not independent of the aforementioned factors, is nevertheless typically of practical intersest and so is also evaluated. Due to limitations of the sparse tensor-vector product support in TensorFlow we explicitly batch the $p'$ index of Equation 9. For computational evaluation we consider a $p'$ batch size of 128 and evaluate the total compute time over all $p'$, i.e. over all $p'$ batches. On an NVIDIA RTX 3090 GPU we observe a wall-clock compute time of $0.0302 \pm 0.0018$, $0.0898 \pm 0.0025$, and $0.3255 \pm 0.0043$ seconds for resolutions of $L = 1024$, $L = 2048$, and $L = 4096$, respectively, when averaged over 10 experiments. With further improvements in sparse tensor support we expect efficiency gains in future.

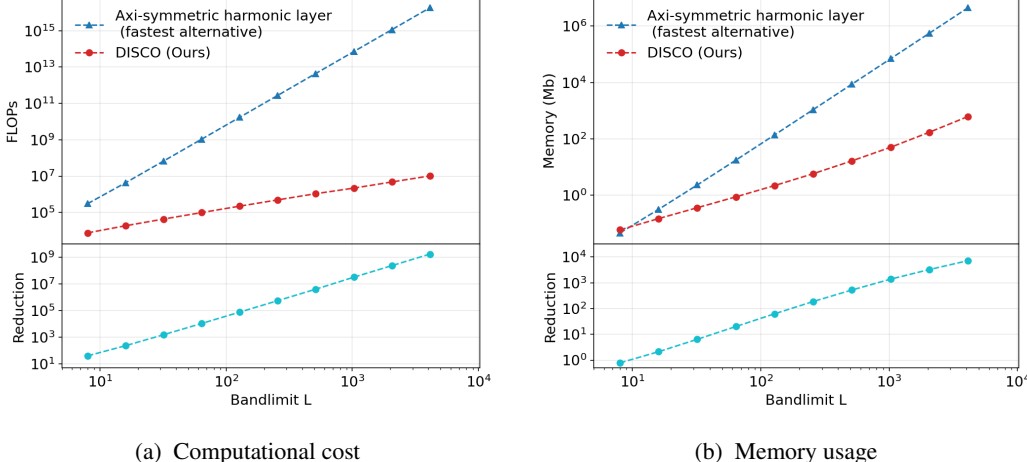

(a) Computational cost  (b) Memory usage

Figure 2: Comparison of computational cost and memory usage of the DISCO spherical convolution and the most efficient alternative equivariant spherical convolution (an axisymmetric harmonic convolution). The DISCO spherical convolution achieves linear scaling in the number of spherical pixels, i.e. $O(L^2)$, for both computational cost and memory usage.

## C  CUSTOM SPARSE GRADIENTS

Since automatic differentiation of sparse tensor operations that include learnable parameters, as in the DISCO convolution, is not supported in either TensorFlow or PyTorch we implement custom sparse gradients.

Consider the sparse tensor representation of the DISCO convolution

$$h_{dj} = \sum_i \Psi_{ji} f_{di}, \tag{17}$$

where, for completeness, we also indicate the data instance by index $d$. Since CNN architectures typically include many convolutional layers, with outputs of one layer providing the inputs for another, to compute gradients of the loss function with respect to the parameters of the model it is necessary to compute the gradient of the convolution with respect to both input and dense filter tensor:

$$\frac{\partial h_{dj}}{\partial f_{d'i'}} = \Psi_{ji'}\delta_{dd'}; \quad \frac{\partial h_{dj}}{\partial \Psi_{j'i'}} = f_{di'}\delta_{jj'}. \tag{18}$$

---

**Algorithm 1** Function to compute custom sparse gradients in TensorFlow.

---

**Input:** The upstream gradient $\dfrac{\partial \sigma}{\partial h_d}$ , input signal $f_d$ and sparse filter tensors $\Psi, \Psi^T$

**Output:** Activation gradient $\left(\dfrac{\partial \sigma}{\partial h}\dfrac{\partial h}{\partial f}\right)_d$ and sparse kernel tensor gradient $\dfrac{\partial \sigma}{\partial h}\dfrac{\partial h}{\partial \Psi}$

1. $\left(\dfrac{\partial \sigma}{\partial h}\dfrac{\partial h}{\partial f}\right)_d = \texttt{tf.sparse.sparse\_dense\_matmul}\left(\Psi^T, \dfrac{\partial \sigma}{\partial h_d}\right)$

2. $i', j' = \Psi.\texttt{indices}$

3. $f'_d = \texttt{tf.gather}(f_d, i')$

4. $\dfrac{\partial \sigma'}{\partial h_d} = \texttt{tf.gather}\left(\dfrac{\partial \sigma}{\partial h_d}, j'\right)$

5. $\dfrac{\partial \sigma}{\partial h}\dfrac{\partial h}{\partial \Psi} = \sum_d \texttt{tf.math.multiply}\left(\dfrac{\partial \sigma'}{\partial h_d}, f'_d\right)$

---

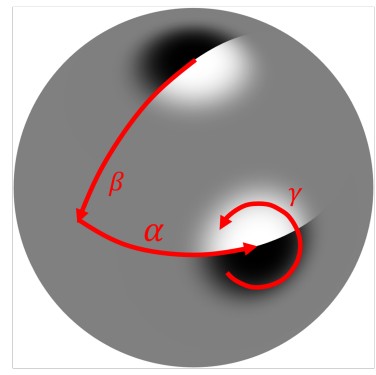

(a) $R = Z(\alpha)Y(\beta)Z(\gamma) \in \mathrm{SO}(3)$ rotation

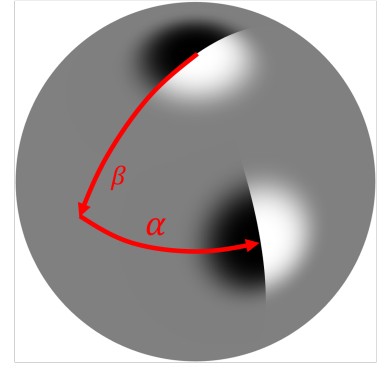

(b) $R = Z(\alpha)Y(\beta) \in \mathrm{SO}(3)/\mathrm{SO}(2)$ rotation

Figure 3: $\mathrm{SO}(3)$ and $\mathrm{SO}(3)/\mathrm{SO}(2)$ rotations.

Pointwise activation functions $\sigma(\cdot)$ are typically included in CNN architectures to introduce non-linearity in an equivariant manner. Gradients can be traced through activation functions by

$$\left(\frac{\partial \sigma}{\partial h}\frac{\partial h}{\partial f}\right)_{d'i'} = \sum_{dj}\frac{\partial \sigma}{\partial h_{dj}}\frac{\partial h_{dj}}{\partial f_{d'i'}} = \sum_{j}\frac{\partial \sigma}{\partial h_{d'j}}\Psi_{ji'}; \tag{19}$$

$$\left(\frac{\partial \sigma}{\partial h}\frac{\partial h}{\partial \Psi}\right)_{j'i'} = \sum_{dj}\frac{\partial \sigma}{\partial h_{dj}}\frac{\partial h_{dj}}{\partial \Psi_{j'i'}} = \sum_{d}\frac{\partial \sigma}{\partial h_{dj'}}f_{di'}. \tag{20}$$

Equation 19 is a sparse-dense tensor multiplication but with the $\Psi$ matrix transposed in input and output coordinates. Thus, we can use a similar method to Section 4.2 to convert to a compressed sparse tensor multiplication. Note also that we need only update the non-zero values of $\Psi_{ji}$, for which we know the corresponding indices. Hence, we need only compute Equation 20 for the sparse set of indices of $j'i'$. Pseudo code of our custom sparse gradient implementation is shown in Algorithm 1.

We validate our implementation of custom sparse gradients presented here against gradients for a dense tensor computed by automatic differentiation (a highly inefficient approach where the dense tensor has many entries set to zero), finding agreement to machine precision.

## D    EQUIVARIANCE TESTS

To test the equivariance of the DISCO spherical convolutions we perform similar equivariance tests to Cobb et al. (2021, Appendix D) but with rotations restricted to $\mathrm{SO}(3)/\mathrm{SO}(2)$ (we provide a visual illustration of $\mathrm{SO}(3)$ and $\mathrm{SO}(3)/\mathrm{SO}(2)$ rotations in Fig. 3). To demonstrate the equivariance of our DISCO layers at high resolution we perform equivariance tests for $L \in \{128, 256, 512, 1024\}$. We consider $N_f = 20$ random spherical signals $\{f_i\}_{i=1}^{N_f}$ with harmonic coefficients sampled from the standard normal distribution and $N_Q = 20$ random rotations $\{Q_j\}_{j=1}^{N_Q}$ sampled uniformly on $\mathrm{SO}(3)/\mathrm{SO}(2)$. Note that for axisymmetric filters, this is equivalent to a random $\mathrm{SO}(3)$ rotation since the first rotation about its axis leaves the filter unchanged. To demonstrate asymptotic $\mathrm{SO}(3)$ equivariance for directional filters (see Section 3.2), we consider rotations of the form $Q_j = Z(\alpha_j)Y(\beta)Z(\gamma_j)$ for random $\alpha_j, \gamma_j$ and constant $\beta \in \{0°, 5°, 10°\}$. We measure the extent to which our DISCO spherical convolution operator $\mathcal{D} : \mathrm{L}^2(S^2) \to \mathrm{L}^2(S^2)$ is equivariant by evaluating the mean relative equivariance error

$$\epsilon(\mathcal{D}(\mathcal{Q}f), \mathcal{Q}(\mathcal{D}(f))) = \frac{1}{N_f}\frac{1}{N_Q}\sum_{i=1}^{N_f}\sum_{j=1}^{N_Q}\frac{\|\mathcal{D}(\mathcal{Q}_j f_i) - \mathcal{Q}_j(\mathcal{D}(f_i))\|}{\|\mathcal{D}(\mathcal{Q}_j f_i)\|} \tag{21}$$

resulting from pre-rotation of the signal, followed by application of $\mathcal{D}$, as opposed to post-rotation after application of $\mathcal{D}$, where the operator norm $\|\cdot\|$ is defined by the inner product on the sphere $\langle\cdot,\cdot\rangle_{\mathrm{L}^2(S^2)}$.

Table 5: Mean relative equivariance error expressed as a percentage (i.e. $100 \times \epsilon$) for axisymmetric and directional filters (best and worst cases are shown, where worst is shown in gray inside brackets).

| Resolution $L$ | Axisymmetric filters | Directional filters | | |
|---|---|---|---|---|
| | | $\beta = 0°$ | $\beta = 5°$ | $\beta = 10°$ |
| 128 | $0.04 \pm 0.01$ $(0.27 \pm 0.02)$ | $0.02 \pm 0.00$ $(0.04 \pm 0.03)$ | $0.86 \pm 0.05$ $(1.11 \pm 0.60)$ | $3.27 \pm 0.15$ $(3.85 \pm 2.20)$ |
| 256 | $0.04 \pm 0.01$ $(0.33 \pm 0.03)$ | $0.01 \pm 0.00$ $(0.02 \pm 0.01)$ | $0.83 \pm 0.02$ $(1.18 \pm 0.72)$ | $3.24 \pm 0.08$ $(4.53 \pm 2.31)$ |
| 512 | $0.04 \pm 0.01$ $(0.48 \pm 0.03)$ | $0.00 \pm 0.00$ $(0.01 \pm 0.01)$ | $0.83 \pm 0.01$ $(1.31 \pm 0.85)$ | $3.23 \pm 0.05$ $(4.30 \pm 2.43)$ |
| 1024 | $0.04 \pm 0.01$ $(0.55 \pm 0.00)$ | $0.00 \pm 0.00$ $(0.01 \pm 0.00)$ | $0.82 \pm 0.00$ $(1.04 \pm 0.69)$ | $3.23 \pm 0.02$ $(3.93 \pm 2.22)$ |

We do not impose filters be bandlimited. As the smoothness of the filter reduces, spherical aliasing is increased, and equivariance errors are increased. We therefore consider best and worst cases to bracket equivariance errors. For the worst case scenario we allow filter parameters to be random and consider $\theta_{\text{cutoff}} = 5\pi/L$, with 4 nodes along $\theta$ and $\phi$. For the best case scenario we use the same nodes and cutoff but consider smoother filters, with the axisymmetric filter given by the smooth function $\psi_{\text{S}}(\theta) = \exp(-\theta_{\text{cutoff}}^2/(\theta_{\text{cutoff}}^2 - \theta^2))$ at node positions and the directional filter by $\psi_{\text{S}}(\theta)\cos(\phi)$. Equivariance errors computed in this manner for the DISCO spherical convolution are presented in Table 5.

For axisymmetric filters we achieve excellent rotational equivariance. For directional filters, for which equivariance is asymptotic since we consider $\text{SO}(3)/\text{SO}(2)$ rotations, we achieve excellent equivariance in the limit of latitudinal rotation $\beta = 0°$. As $\beta$ increases, equivariance errors increase as expected but, nevertheless, we observe only moderate equivariance error of just a few percent at $\beta = 10°$ (this compares favorably to the equivariance error of $\sim 35\%$ for a spherical ReLU at $L = 32$; Cohen et al. 2018; Cobb et al. 2021).

# E  ADDITIONAL INFORMATION ON BENCHMARK EXPERIMENTS

## E.1  MNIST

**Data.** We project the MNIST digits onto the sphere at resolution $L = 1024$, using the same projection as in Cohen et al. (2018). We also standardize the data by subtracting the mean and dividing by the standard deviation (both computed from the entire dataset).

**Architecture.** We consider two models consisting of 7 convolutional layers, one with DISCO spherical convolutions and one with standard planar convolutions for comparison. Convolutional layers are followed by a global integration over the sphere (Cobb et al., 2021) for the spherical model and a global pooling for the planar model, which are followed by 2 dense layers, where all layers include ReLU activations. As typical for CNNs, we continually reduce the resolution of feature maps and increase the number of channels through the model. The bandlimit and number of channels $(L, \text{channels})$ considered for each convolutional layer are as follows when progressing through the model: $(1024, 4)$, $(512, 8)$, $(256, 16)$, $(128, 32)$, $(64, 64)$, $(32, 128)$, $(16, 256)$. For the DISCO model we consider a hybrid model containing DISCO spherical convolutions for early layers that operate at high-resolution and harmonic convolutions for subsequent layers operating at resolutions $L \leq 64$. For both the spherical and planar model we adopt depthwise separable convolutions for channel mixing for layers at resolutions $L \leq 64$ and the standard channel treatment otherwise. For the DISCO spherical convolutions we adopt axisymmetric filters with $\theta_{\text{cutoff}} = 3\pi/L$ and 4 nodes along $\theta$, while for the harmonic convolutions we adopt axisymmetric filters parameterized with 10 nodes in harmonic space, similar to Esteves et al. (2018). For the planar model we adopt $3 \times 3$ convolutional filters. For both models the two proceeding dense layers contain 256 and 10 neurons, respectively. The resulting DISCO model has 449k learnable parameters, while the planar model has 411k. The architectures of the spherical and planar models are closely matched; however, since there are differences in the structure of the spherical and planar layers this results in a slight difference in number of learnable parameters. In any case, this will not induce any difference in the main results of Table 1.

**Training.** We train for 10 epochs usign the ADAM optimizer (Kingma & Ba, 2015), with a learning rate of 0.001 and a batch size of 8.

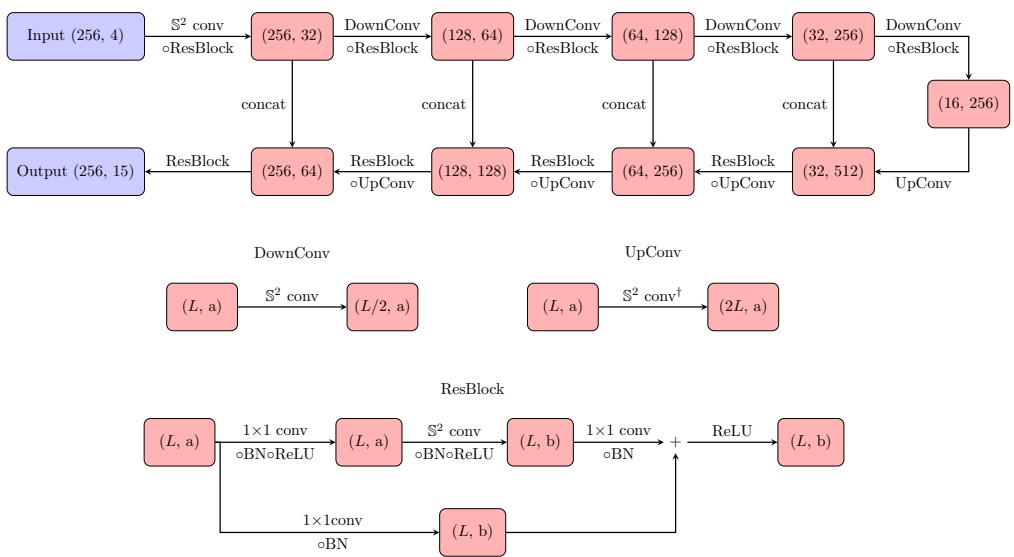

Figure 4: DISCO spherical convolutional residual U-Net architecture.

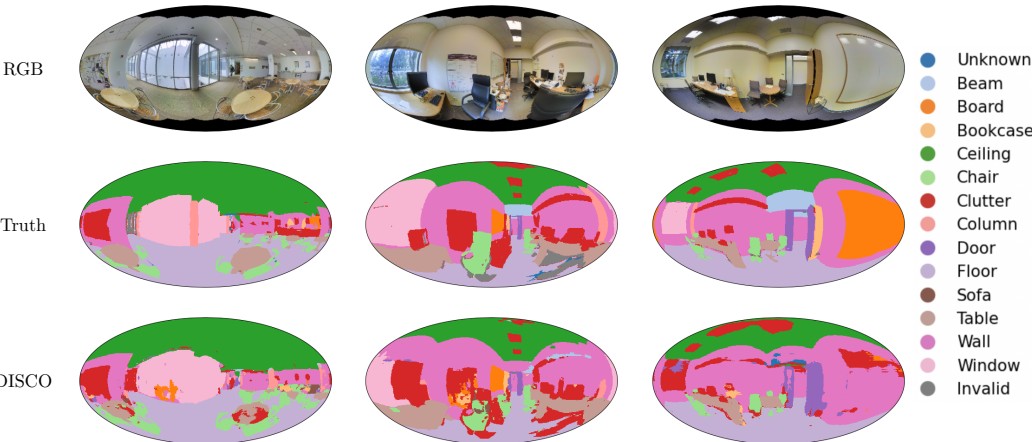

Figure 5: Example predictions for semantic segmentation of 2D3DS data.

## E.2 SEMANTIC SEGMENTATION: 2D3DS

**Data.** The 2D3DS dataset (Armeni et al., 2017) consists of 1413 equirectangular RGB-Depth indoor 360° images, with each pixel belonging to one of 14 classes. We downsample the images to $L = 256$ with bilinear interpolation for the RGB-D images and nearest-neighbor interpolation for the classes. We standardize the RGB-D data by subtracting the mean and dividing by the standard deviation channel-wise. We also experiment with image augmentation via random mirroring, i.e. switching the co-ordinate $\phi \to -\phi$, which improves performance a little (see Table 2).

**Architecture.** We consider a U-Net style architecture with residual blocks as illustrated in Figure 4. Convolutional layers are typically composed of depthwise-separable DISCO spherical convolutions. Downsampling layers are DISCO spherical convolutions but with outputs at half the resolution, i.e. $L/2$. For upsampling layers we use transposed DISCO spherical convolutions sampled at double the resolution, i.e. $2L$. We use spherical batch-normalizations and consider similar residual blocks to Jiang et al. (2019). Note in the final layer we consider 14 channels to represent the different classes and also consider 1 invalid channel. We consider models with axisymmetric (DISCO-Axisymmetric), directional-separable (DISCO-Separable) or unconstrained directional (DISCO-Directional) filters for all convolution layers. For axisymmetric filters we set $\theta_{\text{cutoff}} = 3\pi/L$ and 4 nodes along $\theta$ for $L > 64$, else we use standard harmonic convolutions with 10 nodes in har-

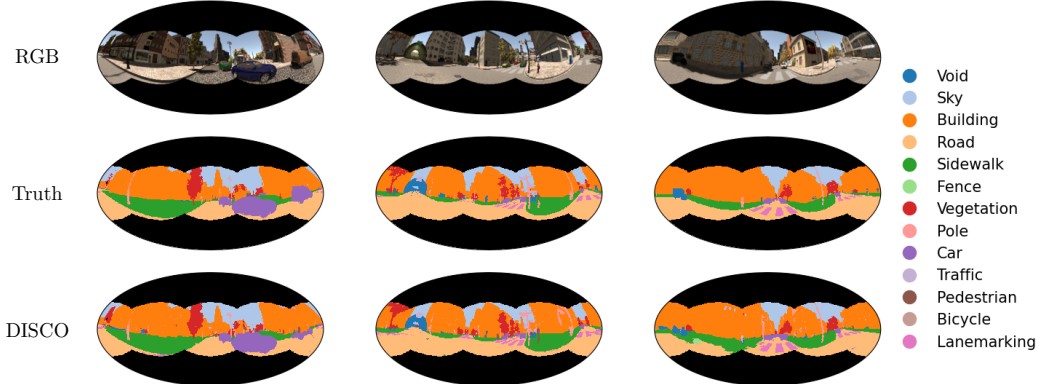

Figure 6: Example predictions for semantic segmentation of Omni-SYNTHIA data.

monic space, similar to Esteves et al. (2018). For separable filters we set $\theta_{\text{cutoff}} = 3\pi/L$ and 4 nodes along $\theta$ and $\phi$. Finally, we also use unconstrained filters where the nodes are a square $3 \times 3$ pixel grid projected onto the sphere. The projection is such that the central node is on the pole at $\theta = 0$ and the edge nodes are at $\theta = \frac{\pi}{L}$, $\phi \in \{0, \pm\frac{\pi}{2}, \pi\}$ and $\theta = \frac{\sqrt{2}\pi}{L}$, $\phi \in \{\pm\frac{\pi}{4}, \pm\frac{3\pi}{4}\}$. We use bilinear interpolation on this regular grid to obtain values outside the nodes. For the DISCO-Directional model we also consider the case with the mirror augmentation discussed above (DISCO-Directional-Aug). The models DISCO-Axisymmetric, DISCO-Separable and DISCO-Directional have 1.78M, 1.47M and 1.68M learnable parameters respectively.

**Training.** We train for 120 epochs using the ADAM optimizer (Kingma & Ba, 2015) with a batch size of 8 and learning rate of 0.01 which is reduced on plateau by 0.1 with patience 8. We use a class-wise weighted cross-entropy loss to balance the class examples. We also weight the loss and metrics by the area of each pixel on the sphere. We use the same 3-fold split for cross-validation as in Jiang et al. (2019). Example segmentation results are shown in Figure 5.

### E.3    SEMANTIC SEGMENTATION: OMNI-SYNTHIA

**Data.** The Omni-SYNTHIA dataset (Ros et al., 2016) consists of 2269 panoramic RGB images of outdoor city sceneries with each pixel belonging to one of 13 classes. We downsample the images to $L = 256$ with bilinear interpolation for the RGB images and nearest-neighbour for the classes. We also experiment with copy-and-paste augmentation (Ghiasi et al., 2021), which is straightforward to implement for our sampling scheme and improves performance a little (see Table 3). This is implemented by choosing a random training image, then choosing a random small object out of pole, traffic sign, pedestrian, or bike and pasting this object on a different training image for both RGB and segmented values. The pasted objects are also translated randomly by $\phi \to n\pi/L$ which can be performed straightforwardly for our sampling scheme; we also do a random mirroring of the pasted object.

**Architecture.** We consider the same backbone of a DISCO spherical convolutional residual U-Net architecture as for the 2D3DS data (as in Figure 4) but with 13 output channels (+1 channel for invalid) and 3 channels for the input. Here we consider only separable directional filters parameterized identically as for 2D3DS (DISCO-Separable). We also consider the case with the copy-and-paste augmentation discussed above (DISCO-Separable-Aug). The resulting model has 1.47M learnable parameters.

**Training.** We train for 70 epochs using the ADAM optimizer Kingma & Ba (2015) with a batch size of 8 and learning rate of 0.01, which is reduced on plateau by 0.1 with patience 8. We use a class-wise weighted cross-entropy loss to balance the class examples. We also weight the loss and metrics by the area of each pixel on the sphere. We use the same training/test split as in Zhang et al. (2019). Example segmentation results are shown in Figure 6.

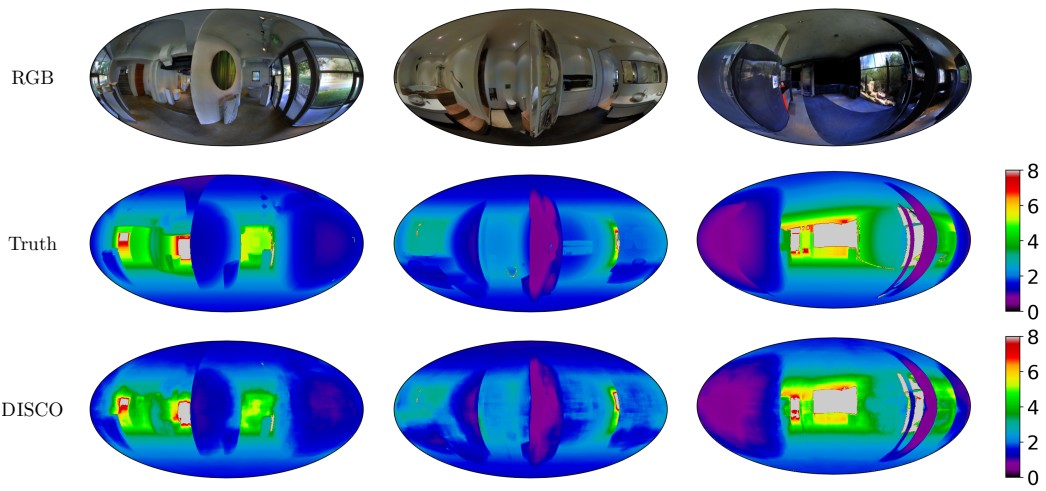

Figure 7: Example predictions for depth estimation of Pano3D data (depth plotted in meters).

## E.4 DEPTH ESTIMATION: PANO3D ON MATTERPORT3D

**Data.** The Matterport3D dataset (Chang et al., 2017) contains 7907 spherical RGB images from which we predict spherical depth maps. We downsample the RGB-Depth images to $L = 512$ with bilinear interpolation.

**Architecture.** We consider the same backbone of a DISCO spherical convolutional residual U-Net architecture as for the semantic segmentation problem but adapted for depth estimation. The architecture is similar to Figure 4 but we replace each ResBlocks simply by a DISCO spherical convolution. All resolutions are doubled to start at resolution $L = 512$ for the input. We consider 3 input channels and 1 output channel (depth). The number of channels for the down leg of the model are $(64, 64, 128, 256, 512)$, which is reversed for the up leg. For the upsampling layers we increase the bandlimit by 2 using nearest neighbor interpolation. We adopt this upsampling rather than transposed convolutions to simplify transfer learning (discussed below) from planar to spherical convolutions. Due to the higher resolution we use smaller batch sizes, which means batch-normalization is unsuitable and so we therefore use group-normalization (Wu & He, 2018) with 16 groups. Here we apply a group-normalization and ReLU after every DISCO spherical convolution. We consider only unconstrained directional filters (DISCO-Directional), adopting the same parameterization as for the segmentation problems, as described above. The resulting model has 658k learnable parameters.

**Training.** We use an area weighted $\ell_1$ loss and the same area weighted metrics as in Albanis et al. (2021). We use the same train/test/validation split as in Albanis et al. (2021). For faster training we use transfer learning by first training the above architecture with planar convolution layers with $3 \times 3$ filters, for 60 epochs using the ADAM optimizer (Kingma & Ba, 2015), with a batch size of 2 and learning rate of 0.0001. The $3 \times 3$ filter weights are then easily transferred to the filters projected on the sphere for the unconstrained directional DISCO filters. After the weight transfer, the DISCO model is then trained for a further 60 epochs with the same hyperparameters. Example depth estimation results are shown in Figure 7.

