# OpenReview forum: "Scalable and Equivariant Spherical CNNs by Discrete-Continuous (DISCO) Convolutions"
_ICLR.cc/2023/Conference — ICLR 2023 poster_

### Official Review · Reviewer_kRdr · 2022-10-23

**Confidence:** 3
**Correctness:** 4
**Technical Novelty And Significance:** 4
**Empirical Novelty And Significance:** 4
**Recommendation:** 8

**Clarity, Quality, Novelty And Reproducibility:**

I think the paper has several novel ideas for making group convolutions scalable while retaining equivariance, describing how these ideas are applicable for spherical signals and showing large computation and memory improvements. The paper is generally clear on most of these aspects. I have difficulty in understanding some parts which I have addressed under weaknesses. In principle, it may be reproducible, but I think, practically, it would be quite difficult, especially, writing custom sparse gradients.

**Details Of Ethics Concerns:**

No ethical concerns.

**Strength And Weaknesses:**

Strengths:

1. The paper addresses an important question which is how to make group convolution scalable while maintaining equivariance. Earlier attempts at scalability required full discretization which sacrificed equivariance. In contrast, here, only the band-limited signal is discretized to a level where all the information is still maintained, and the filters and the outputs are continuous. The authors then show that this maintains equivariance while making the computation much cheaper as expensive Fourier transforms are no longer required. This is a novel direction.

2. These general ideas are made useful in practice for spherical images by presenting DISCO version of the spherical and $SO(3)$ convolution.

2. The theoretical speedup is huge and the computation is linear in the number of pixels when the filters have local support and are small.

3. The authors describe the implementation in detail using sparse tensor which shows linear scaling with number of pixels. This is very good result, in my opinion. And as the authors say, for 4k images, this shows a theoretical speedup of $10^9$.

4. The authors also describe smart tricks for memory optimization by exploiting the patterns in the filter arrays and the convolution operation leading to a $10^4$ improvement.

5. Experimental results on equivariance tests clearly show that the method maintains nearly perfect equivariance. The error mainly comes from having to eventually discretize the integral in order to compute the convolution on a computer.

6. Although I am not familiar with the baselines, results on semantic segmentation and depth estimation datasets show very good results using DISCO. I am especially happy to see the results on depth estimation performing comparably with a U-Net that uses 40x the number of parameters.

Weaknesses:

1. There is a crucial detail that I am not following in the method which I hope the authors can explain. I understand that the sampling theorems on a compact group or its homogeneous space (e,g, sphere) show how many samples are needed to preserve all the information after discretization. But I don't see how this leads to Eq. (3) (and Eq(4)), Given a discrete signal, I would think we would still need to first reconstruct the underlying continuous signals by some low-pass filter (convolving with some form of a sinc function?) on the space for performing the convolution. I am not sure I understand how the result in Eq. (3) and Eq. (4) is possible.

2. While the speedup and memory improvements are fantastic, they are still based on theoretical analysis. Can the authors show computation times observed empirically?

3. Experimental results should ideally have error bars and statistical significance results.

4. I urge the authors to try and release their software implementation to make the experiments easily reproducible, but of course, this is not required.

**Summary Of The Paper:**

The paper proposes a novel form of group convolution called Discrete-Continuous (DISCO) group convolution, which is targeted at making group convolutions scalable and at the same time retain equivariance properties. DISCO discretizes the input signal space while the filters and output space are maintained to be continuous. If the input signals are band-limited and there are sufficient samples for reconstruction, then the group convolution can be performed exactly using DISCO convolution. This means that the computation is much faster and also, the operation is exactly equivariant to group action. The authors then discuss the special case of signals on the $S^2$ sphere with the natural $SO(3)$ action as well as $SO(3)/SO(2)$ transformation. They also discuss in detail how to build a computationally scalable DISCO convolution using sparse tensors and other memory optimization. Experiments are performed on Rotated-MNIST recognition and semantic segmentation and depth estimation for omnidirectional images.

**Summary Of The Review:**

I think this paper has several very good ideas, implementation and experimental results. There are some crucial details that I am not able to understand by reading the paper which I hope the authors can explain. For now, I am giving this paper a "marginally above threshold", but would be more than happy to increase my rating to an accept or a strong accept depending on how the authors address the weaknesses I have listed.

UPDATE AFTER AUTHOR RESPONSE:

I thank the authors for their explanations and improvements to the paper. Based on this and the other reviews and response, I am recommending acceptance of the paper.

---

> ### Author Response · Authors · 2022-11-18
> **Response to Reviewer kRdr (1 of 2)**
>
> We thank the Reviewer for their comments.  We respond to each comment in turn below.  The Reviewer's original comments are italicized, while our responses are given in Roman font.  All revisions to the manuscript are highlighted in red.
>
> *There is a crucial detail that I am not following in the method which I hope the authors can explain. I understand that the sampling theorems on a compact group or its homogeneous space (e,g, sphere) show how many samples are needed to preserve all the information after discretization. But I don't see how this leads to Eq. (3) (and Eq(4)), Given a discrete signal, I would think we would still need to first reconstruct the underlying continuous signals by some low-pass filter (convolving with some form of a sinc function?) on the space for performing the convolution. I am not sure I understand how the result in Eq. (3) and Eq. (4) is possible.*
>
> The Reviewer is quite right in appreciating there is a great deal of subtlety here.  We have added a new Appendix A to more thoroughly discuss the connections between sampling theorems and quadrature on the sphere, which we hope helps to add clarity.  Furthermore, we also present the explicit quadrature rule for the antipodal sampling of the McEwen & Wiaux (2011) sampling theorem that we adopt, which we derived as part of this work.  The result is very similar to the quadrature for the standard McEwen & Wiaux (2011) sampling but is included for clarity and reproducibility (since it has not appeared in the literature previously).
>
> Coming to the specific question raised by the Reviewer, the situation is much like in the Euclidean setting.  If the continuous signal is bandlimited at $L$ then the sampling scheme associated with the adopted sampling theorem ensures sufficient samples to fully capture all information content of the signal.  If, on the other hand, the signal is not bandlimited at $L$, then the resolution of the sampling scheme is insufficient to capture all information content and aliasing occurs.  The situation here is analogous to the Euclidean setting: if a signal is not Nyquist sampled in the Euclidean setting then the resolution of the sampled signal will not be sufficient to capture all information content and aliasing will occur.
>
> In the setting where a signal is not bandlimited, or bandlimited at a much higher degree than supported by the resolution of the sampling scheme, then the signal could in principle be low pass filtered as the Reviewer comments to limit the bandlimit.  Again, the process would be analogous to the Euclidean setting (and is something that would typically be considered in analog to digital conversion).  For the purposes of our work we consider bandlimited signals on the sphere and hence the sampling scheme corresponding to a sampling theorem is sufficient to capture all information content of the signal.

---

> > ### Author Response · Authors · 2022-11-18
> > **Response to Reviewer kRdr (2 of 2)**
> >
> > *While the speedup and memory improvements are fantastic, they are still based on theoretical analysis. Can the authors show computation times observed empirically?*
> >
> > The computational cost results presented in Appendix B [previously Appendix A] are indeed theoretical FLOP counts and do not account for practical considerations such as data communication, memory management and throughput.  The advantage of this presentation is that we can abstract away details of a particular implementation and its level of optimization.  Moreover, we are then also agnostic to the selection of hardware used.  Nevertheless, we appreciate the Reviewer's comment that a practical computational evaluation would be of interest.  We have therefore performed additional practical computational time experiments and added them to Appendix B [previously Appendix A].  Furthermore, we note that the memory usage evaluation is already empirical and not purely theoretical since we explicitly construct the sparse tensor representation of the DISCO spherical convolution and calculate the memory required for both forward and backward passes.  We have also clarified this in the manuscript.
> >
> > *Experimental results should ideally have error bars and statistical significance results.*
> >
> > We agree with the Reviewer that this would be interesting but we unfortunately do not have the computational resources to repeat all experiments numerous times.  While the methods we present are computationally efficient they are nevertheless still quite computationally demanding.  We have repeated experiments where possible in order to include errors bars, e.g. in equivariance tests and wall-clock compute timing measurements.
> >
> > *I urge the authors to try and release their software implementation to make the experiments easily reproducible, but of course, this is not required.*
> >
> > We intend to make our code implementing the methods discussed in the paper public.  It was always our intention to make this code public, although commercial considerations prevented us for making the code public previously.  We plan to make the code public on the publication of this article.
> >
> > *In principle, it may be reproducible, but I think, practically, it would be quite difficult, especially, writing custom sparse gradients.*
> >
> > As commented above, we will make the code public on publication of the manuscript.  We hope adding further details regarding the connection between sampling theorems and quadrature on the sphere, as discussed above, will also help to improve reproducibility.  In response to another Reviewer's comment we have also added pseudo code for the custom sparse gradient implementation to further aid reproducibility.

---

### Official Review · Reviewer_ef9H · 2022-10-26

**Confidence:** 4
**Correctness:** 4
**Technical Novelty And Significance:** 3
**Empirical Novelty And Significance:** 4
**Recommendation:** 8

**Clarity, Quality, Novelty And Reproducibility:**

Clarity and quality: The paper is presented impeccably -- I quite enjoyed reading it. The development of the main contribution of the paper is done in a way that makes it seem obvious.

Novelty: The paper presents a heavily engineered spherical CNN that can scale linearly with number of pixels. It does a clever combination of continuous and discrete aspects in the convolution formula, using an interesting sampling theorem. The results on dense prediction tasks for spherical images are also interesting and new.

Reproducibility: Given the writing and presentation of the paper, I am confident that it is easy to reproduce. But I am a little turned off by non-availability, and lack of indication thereof, of the code and experimental replication.

Minor comment:
- On page 2, f, \psi: G/N -> \mathbb{R} should be G/H

**Strength And Weaknesses:**

- The paper is very well motivated and solves an important problem -- that is of scaling spherical CNNs so that they become useful for larger images and for dense prediction.
- The main conceptual contribution is quite simple, but still significant.
- The system is implemented very efficiently, and the engineering effort and attention to detail is admirable.
- The dense prediction results are good, and unlike other spherical CNNs.
- Paper is also very well written.

Weaknesses
- The paper seems to make no indication about code availability. I will definitely consider this to be a negative.

**Summary Of The Paper:**

There are now a wide crop of Spherical CNN models, which can be classified along several different axes e.g. fully Fourier/fully real/partly real and partly harmonic; or continuous models and discrete models. If we focus on the latter, existing continuous models guarantee rotational equivariance, while being computationally expensive (and this varies depending on how much equivariance error is tolerated). On the other hand, discrete approaches (such as those that represent spherical signals as a graph) are more computationally attractive, but don't handle equivariance in a satisfactory manner. Part of the expense of the continuous approaches comes from the prohibitive cost of the spherical/rotational Fourier transform. This paper proposes a framework that attempts to combine the advantages of both of these styles of models, leading to the proposed spherical CNN scaling up much better than existing models, and also becoming capable of doing dense prediction.

The paper first describes the basic idea of the continuous approach (exemplified by Cohen et al, Kondor et al., Esteves et al. and Cobb et al.), while describing some of the issues concerning sampling and the computational load. It then briefly discusses some of the discrete models (which tend to be graph based and are inherently limited in the symmetries that they can capture). It is also mentioned that recent scattering-based methods still don't manage to work quite well with high-resolution images -- something that I am in full agreement with. The approach of the paper is described, at a high-level, in sectoin 3.1. The key equations are equations 3 and 4. The main innovation of this section is to replace the integral by a quadrature (possible source of error), which is evaluated at some sample positions. However, the fllter, and its transformation is kept continuous -- and consequently does not lead to errors. The quadrature error is minimized (or in principle can be avoided) by appeal to an interesting sampling theorem from a decade ago. After this the authors describe a few area where the computational load can be further chopped off (and permit linear scaling with the pixels). This includes reducing the space of rotations SO(3) to SO(3)/SO(2) (the authors keep using equivariance to SO(3)/SO(2) -- which I am not sure what it means since it is not a group, it is quotient space). Filter parameterization also uses simple linear interpolation after sample points are selected and reduces the load a bit further (although possibly being a source of error). The authors also detail how they leverage the sparse representations that are obtained for further computational (and memory) gains. The memory gains are also augmented by making use of a symmetry in the sampling procedure. Eventually, the authors also describe an extremely well-engineered system that is able to scale to levels unlike other spherical CNNs and performs wells on some experiments (including dense segmentation).

**Summary Of The Review:**

See above.

---

> ### Author Response · Authors · 2022-11-18
> **Response to Reviewer ef9H (1 of 1)**
>
> We thank the Reviewer for their comments.  We respond to each comment in turn below.  The Reviewer's original comments are italicized, while our responses are given in Roman font.  All revisions to the manuscript are highlighted in red.
>
> *(the authors keep using equivariance to SO(3)/SO(2) -- which I am not sure what it means since it is not a group, it is quotient space).*
>
> This is a fair point.  We do indeed make a small abuse of terminology here since, as the Reviewer rightly points out, the quotient space $SO(3)/SO(2)$ is not a group.  What we mean is that the commutativity relation for the transform $\mathcal{Q}$ holds, i.e. $((\mathcal{Q}f) \star \psi)(R) \rightarrow (\mathcal{Q}(f \star \psi))(R)$ for $Q,R\in \text{SO(3)/SO(2)}$. To avoid adopting a cumbersome terminology we nevertheless keep the current terminology but comment on the overloading of terminology in the manuscript.
>
> *The paper seems to make no indication about code availability. I will definitely consider this to be a negative.*
>
> We intend to make our code implementing the methods discussed in the paper public.  It was always our intention to make this code public, although commercial considerations prevented us for making the code public previously.  We plan to make the code public on the publication of this article.
>
> *The paper is presented impeccably -- I quite enjoyed reading it.  The development of the main contribution of the paper is done in a way that makes it seem obvious.*
>
> We thank the Reviewer very much for their kind comment and are delighted to learn they enjoyed reading the manuscript.
>
> *Given the writing and presentation of the paper, I am confident that it is easy to reproduce. But I am a little turned off by non-availability, and lack of indication thereof, of the code and experimental replication.*
>
> As commented above, we will make the code public on publication of the manuscript.  In response to another Reviewer's comment we have also added pseudo code for the custom sparse gradient implementation to further aid reproducibility.  In response to further Reviewer's comment we have added a new Appendix A discussing the connection between sampling theorems and quadrature on the sphere, and derive the explicit form of the quadrature used, which we hope will further improve reproducibility.
>
> *On page 2, $f, \psi: G/N -> \mathbb{R}$ should be $G/H$*
>
> We thank the Reviewer for catching this typo.  This has now been corrected.

---

> > ### Comment · Reviewer_ef9H · 2022-11-18
> > **Thanks**
> >
> > Thanks for your detailed response. I went over the other responses as well. I wanted to write a comment acknowledging the rebuttal. It is good to know that the code will become available.

---

> > > ### Author Response · Authors · 2022-12-11
> > > **Response to Reviewer ef9H (second round)**
> > >
> > > We thank the Reviewer for their further comments.

---

### Official Review · Reviewer_vRza · 2022-11-01

**Confidence:** 4
**Clarity, Quality, Novelty And Reproducibility:** See the comments above.
**Correctness:** 2
**Technical Novelty And Significance:** 2
**Empirical Novelty And Significance:** 2
**Recommendation:** 5

**Strength And Weaknesses:**

Strength:
1. This method combines the advantages of continuous and discrete spherical convolutions, and is both rotation-equivariant and scalable.

Weaknesses:
1. This paper contains many long sentences, reads rather awkwardly and even has many fault wordings, e.g.,
"the sphere is a homogeneous space with global symmetries on which act elements of the group of 3D rotations SO(3)"
"The DISCO approximation of Equation 3 is highly accurate for bandlimited signals, which real-world signals can be well approximately by for sufficient bandlimit."
"The filter is represented continuously so also does not introduce any error."
I think the writing should be significantly improved.

2. To my understanding, the main idea is in Equation (3), i.e., applying continuous filters \psi to the discrete inputs f, and then implementing group convolutions, which is rather trivial.

3.  Restricting rotations to SO(3)/SO(2) will degrade the equivariance over arbitrary 3D rotations to that over z-rotations. Actually, the equivariance w.r.t z-rotations can be easily achieved by employing oriented-aware spherical CNNs, such as HexUNet.

4. The baselines of spheical MNIST and Pano3D are both planar CNNs, which are very weak. The result for 2D3DS is also not the state-of-the-art, as the performance is much weaker than [1].

5. The paper claims that this method is both memory- and computation-efficient. However, there is no practical experimental evaluation about that.

[1] Eder et al. Tangent Images for Mitigating Spherical Distortion. CVPR 2020.

**Summary Of The Paper:**

This paper proposes a kind of scalable and equivariant spherical convolution by applying continuous filters to discrete spherical inputs. The authors claim that this method is more computation- and memory-efficient than previous works.

**Summary Of The Review:**

This idea is trivial and the writing should be significantly improved. Also, their experiments cannot support their claims well.

---

> ### Author Response · Authors · 2022-11-18
> **Response to Reviewer vRza (1 of 3)**
>
> We thank the Reviewer for their comments.  We respond to each comment in turn below.  The Reviewer's original comments are italicized, while our responses are given in Roman font.  All revisions to the manuscript are highlighted in red.
>
> *This paper contains many long sentences, reads rather awkwardly and even has many fault wordings, e.g., "the sphere is a homogeneous space with global symmetries on which act elements of the group of 3D rotations SO(3)" "The DISCO approximation of Equation 3 is highly accurate for bandlimited signals, which real-world signals can be well approximately by for sufficient bandlimit." "The filter is represented continuously so also does not introduce any error." I think the writing should be significantly improved.*
>
> We are sorry to hear the Reviewer found the paper awkward to read at times.  We have revised all of these sentences and generally tried to improve readability further.  Although these longer sentence may have been difficult to read, we believe all were grammatically correct.  We note that other Reviewers found the paper "well written" and "presented impeccably".
>
> *To my understanding, the main idea is in Equation (3), i.e., applying continuous filters \psi to the discrete inputs f, and then implementing group convolutions, which is rather trivial.*
>
> While the general idea of the DISCO convolution may naively appear trivial, there is considerable subtlety. Indeed, as Reviewer ef9H notes: "The development of the main contribution of the paper is done in a way that makes it seem obvious."  The adoption of sampling theory on the sphere is important to avoid any discretization error due to sampling the signal and to minimize approximation error in the computation of the integral by adopting appropriate quadrature weights.  We have added a new Appendix A discussing the connection between sampling theorems and quadrature on the sphere, and derived the explicit form of the quadrature used, which we hope will further clarify our approach.  In any case, irrespective of the triviality (or otherwise) of the approach, a critical concern is the impact that such an approach can have.  We show that our proposed approach provides considerable saving in compute and memory requirements, while also achieving SOTA on numerous dense-prediction benchmark problems.

---

> > ### Author Response · Authors · 2022-11-18
> > **Response to Reviewer vRza (2 of 3)**
> >
> > *Restricting rotations to SO(3)/SO(2) will degrade the equivariance over arbitrary 3D rotations to that over z-rotations. Actually, the equivariance w.r.t z-rotations can be easily achieved by employing oriented-aware spherical CNNs, such as HexUNet.*
> >
> > Full SO(3) equivariance can be captured by the DISCO approach in two ways.
> >
> > For the first approach, directional filters can be rotated by the usual $SO(3)$ rotations.  In this setting full $SO(3)$ rotational equivariance is achieved.  This is discussed in Section 3.3 under the subheading **$SO(3)$ rotational equivariance**.  However, we do not implement this approach due to the computational cost of $SO(3)$ convolutions.  In this scenarios not only is the cost of $S^2 \rightarrow SO(3)$ convolution increased but convolutions of subsequent layers must be performed on $SO(3)$.  While efficient DISCO convolutions on $SO(3)$ can certainly be considered (in an analogous manner to Cohen et al. 2018 and Cobb et al. 2021) and will satisfy $SO(3)$ rotational equivariance, computational cost will be further increased.  In the current manuscript we focus on highly computationally scalable approaches hence we do not implement this approach or consider it further.
> >
> > For the second approach to achieve full SO(3) equivariance, axisymmetric filters may be considered.  In this setting, since the filters are invariant to rotations about their own axis, spherical convolutions with $SO(3)$ and $SO(3)/SO(2)$ rotations are equivalent and so full $SO(3)$ rotational equivariance is achieved.  This is discussed in Section 3.3 under the subheading **Axisymmetric filters and $SO(3)$ rotational equivariance**.  Since the axisymmetric spherical convolution is efficient and outputs remain on $S^2$, we do implement this approach in the DISCO framework.  Indeed, we perform numerical experiments that show that excellent $SO(3)$ rotational equivarance is achieved in this setting, with equivariance errors at the level of $\sim 10^{-4}$ for single precision arithmetic (see Section 5.1 and Appendix D [previously Appendix C]).  Furthermore, for the MNIST experiments where full $SO(3)$ rotational equivariance is desired we adopt axisymmetric filters and demonstrate excellent rotational invariance of the complete architecture (see Section 5.2 and Appendix E.1 [previously Appendix D.1]).  Axisymmetric filters are clearly less expressive than directional filters, nevertheless we and others (Esteves et al. 2018) both show axisymmetric filters can be highly effective.
> >
> > Finally, it one wishes to achieve the expressivity of directional filter kernels, with maximal computational efficiency, then directional filters with $SO(3)/SO(2)$ rotations may be considered, which we show leads to asymptotic $SO(3)/SO(2)$ equivariance.  We stress that this setting is not essential and full $SO(3)$ equivariance can be achieved using either of the approaches discussed above if desired.  Nevertheless, $SO(3)/SO(2)$ equivariance can be desirable in many practical applications since content in spherical signals is often orientated and similar content often appears at similar latitudes, particularly for $360{}^\circ$ panoramic photos and video.  We find in numerical experiments that asymptotic $SO(3)/SO(2)$ equivariance, coupled with the expressivity of directional filter kernels, performs very well in benchmark problems.
> >
> > Furthermore, just as for the HexUNet approach commented by the Reviewer, we achieve exact equivariance with respect to Z rotations. This setting is recovered for $Z(\alpha)Y(\beta)$ when $\beta = 0$ and as we comment in the manuscript there is no restriction on $\alpha$.  We have added a comment to explicitly state that full equivariance with respect to Z rotations is achieved by our approach.

---

> > > ### Author Response · Authors · 2022-11-18
> > > **Response to Reviewer vRza (3 of 3)**
> > >
> > > *The baselines of spheical MNIST and Pano3D are both planar CNNs, which are very weak.*
> > >
> > > For MNIST, the purpose of the experiment is to demonstrate that classification performance remains roughly constant across the various experimental settings for our DISCO model (i.e. NR/NR, R/R, NR/R).  The final setting (NR/R), where training data is not rotated but test data is rotated, relies heavily on the $SO(3)$ rotational invariance of the model.  Since the planar architecture does not exhibit $SO(3)$ rotational invariance, classification fails catastrophically for the planar model in the NR/R setting.  As far as we are aware no other architectures that achieve rotational equivariance are able to run at this resolution ($L=1024$), hence there are no other cases to which we can directly compare.  And in any case, the purpose of this experiment is not achieve the highest classification performance possible but to demonstrate the invariance of the DISCO model.
> > >
> > > Pano3D is a state-of-the-art benchmark that was only recently proposed.  Consequently, no further models have been applied to this benchmark hence there are no other cases to which we can compare, as we comment in the current manuscript.  Nevertheless, extensive effort has been applied to develop an effective planar model for this benchmark, to which we do compare.  It is encouraging to see that DISCO outperforms this model while also using approximately $40 \times$ fewer parameters.  We expect Pano3D will become a common benchmark in the near future, in which case it will then be possible to compare our DISCO architecture to numerous other models.
> > >
> > > *The result for 2D3DS is also not the state-of-the-art, as the performance is much weaker than [1] (Eder et al.).*
> > >
> > > Thank you for bringing this paper to our attention.  We have included results from  Eder et al. (2021) in the results tables for 2D3DS and Omni-SYNTHIA (Table 2 and Table 3).  We noticed that experiments with this method are performed at various resolutions for both benchmark problems.  For a fair comparison results should clearly be compared for experiments at the same resolution.  We have therefore appended to these tables the effective resolution of the experiment $\tilde{L}$.  When considering the same resolution, DISCO still achieves SOTA on both 2D3DS and Omni-SYNTHIA.
> > >
> > > *The paper claims that this method is both memory- and computation-efficient. However, there is no practical experimental evaluation about that.*
> > >
> > > The computational cost results presented in Appendix B [previously Appendix A] are indeed theoretical FLOP counts and do not account for practical considerations such as data communication, memory management and throughput.  The advantage of this presentation is that we can abstract away details of a particular implementation and its level of optimization.  Moreover, we are then also agnostic to the selection of hardware used.  Nevertheless, we appreciate the Reviewer's comment that a practical computational evaluation would be of interest.  We have therefore performed additional practical computational time experiments and added them to Appendix B [previously Appendix A].  Furthermore, we note that the memory usage evaluation is already empirical and not purely theoretical since we explicitly construct the sparse tensor representation of the DISCO spherical convolution and calculate the memory required for both forward and backward passes.  We have also clarified this in the manuscript.

---

> > > > ### Comment · Reviewer_vRza · 2022-11-20
> > > > **Thank you for detailed response, but some issues still remain.**
> > > >
> > > > Thank the authors for detailed response and improvement. However, some issues remain as follows:
> > > > 1. The first issue is about the noveltiy of this work. I understand that you leverage sample theory on the sphere to avoid any discretization error. However, the contribution seems incremental as the sample theory has been widely used in previous works. Also, I am curious that whether using sample theory is necessary, noting some previous works can use Gaussian kernels to achieve equivariance well with equally O(N) computational complexity, such as [1] in the 2D case, [2] in the 3D case and [3] for spherical data.
> > > >
> > > > 2. I agree that SO(3)/SO(2)-equivariance is practical for some applications, but the SO(3)/SO(2) equivariance is indeed much weaker than SO(3)-equivariance, and cannot address more complicated problems, such as molecular property prediction. In addition, can you describe axisymmetric filters more explicitly? Are they equivalent to isotropic filters? In my understanding, if we use isotropic filters to deal with the spherical data, the SO(3)-equivariance can be naturally achieved. I think that some claims about SO(3)/SO(2) are confusing and easy to cause misleading, and it should be claimed more explicitly.
> > > >
> > > > 3. I understand that the experiments on Spherical MNIST is to verify the equivariance of your method. However, some previous methods, such as [4], can also perform very well on this task even with a relatively lower resolution. Results in this paper are weaker than theirs, which cannot reveal the advantages.
> > > >
> > > > 4. Noting that one advantage of DISCO is addressing the spherical data in a high resolution, can you achieve a new SOTA result when applying a higher resolution, just like that in Eder et al.?
> > > >
> > > > [1] Jenner and Weiler. Steerable partial differential operators for equivariant neural networks. ICLR, 2021.
> > > >
> > > > [2] Weiler et al. 3D steerable CNNs: Learning rotationally equivariant features in volumetric data[J]. NeurIPS, 2018.
> > > >
> > > > [3] https://openreview.net/pdf?id=HJeYSxHFDS
> > > >
> > > > [4] Esteves C, Spin-weighted spherical CNNs. NeurIPS, 2020.

---

> > > > > ### Author Response · Authors · 2022-12-11
> > > > > **Response to Reviewer vRza (second round)**
> > > > >
> > > > > We thank the Reviewer for their further comments.  We respond to each comment in turn below.  The Reviewer's original comments are italicized, while our responses are given in Roman font.
> > > > >
> > > > > *The first issue is about the noveltiy of this work. I understand that you leverage sample theory on the sphere to avoid any discretization error. However, the contribution seems incremental as the sample theory has been widely used in previous works. Also, I am curious that whether using sample theory is necessary, noting some previous works can use Gaussian kernels to achieve equivariance well with equally O(N) computational complexity, such as [1] in the 2D case, [2] in the 3D case and [3] for spherical data.*
> > > > >
> > > > > Regarding the work targeting the spherical setting, i.e. [3], fully SO(3) rotational equivariance is not achieved, contrary to the Reviewer's comment.  While the gauge equivariant CNNs presented in [3] are a seminal contribution, the icosahedtral (spherical) CNNs are only equivariant to a set of six orientations.  Indeed, as commented in the article "None of the models generalize to continuously rotated inputs..." [3].
> > > > >
> > > > > *I agree that SO(3)/SO(2)-equivariance is practical for some applications, but the SO(3)/SO(2) equivariance is indeed much weaker than SO(3)-equivariance, and cannot address more complicated problems, such as molecular property prediction. In addition, can you describe axisymmetric filters more explicitly? Are they equivalent to isotropic filters? In my understanding, if we use isotropic filters to deal with the spherical data, the SO(3)-equivariance can be naturally achieved. I think that some claims about SO(3)/SO(2) are confusing and easy to cause misleading, and it should be claimed more explicitly.*
> > > > >
> > > > > We have studied this again and now realize that the spherical convolution with SO(3)/SO(2) rotations in fact satisfies a form of asymptotic SO(3) equivariance (rather than asymptotic SO(3)/SO(2) equivariance), which we hadn't appreciated previously.  This result is stronger than asymptotic SO(3)/SO(2) equivariance but it weaker than full SO(3) since it remains asymptotic (as $\beta \rightarrow 0$).  In any case, as commented previously, full SO(3) rotational equivariance can be achieved by adopting axisymmetric filters.  Axisymmetric filters are filters that don't have directional structure and so are invariant about the axis on which they are centered (typically the z axis).
> > > > >
> > > > > *I understand that the experiments on Spherical MNIST is to verify the equivariance of your method. However, some previous methods, such as [4], can also perform very well on this task even with a relatively lower resolution. Results in this paper are weaker than theirs, which cannot reveal the advantages.*
> > > > >
> > > > > We did not optimise the hyperparameters of the spherical MNIST experiments.  Since others have not performed high-resolution MNIST experiments we did not feel the need to optimise the model to achieve the best accuracy possible.  Instead we simply compared to a planar network to show that our DISCO approach exhibits excellent equivariance. There are indeed many works that achieve higher accuracy on low resolution MNIST experiments but these should not be directly compared since the data-sets are different (low-resolution compared to high-resolution) and our DISCO model has not been optimised for this problem.
> > > > >
> > > > > *Noting that one advantage of DISCO is addressing the spherical data in a high resolution, can you achieve a new SOTA result when applying a higher resolution, just like that in Eder et al.?*
> > > > >
> > > > > In terms of further high-resolution experiments, while we agree these would be interesting, we unfortunately do not have the computational resources to run these experiments during this further discussion period of the review process.  And in any case, we understand that further revisions to the manuscript should not be made at this stage.

---

### Official Review · Reviewer_D1fT · 2022-11-03

**Confidence:** 4
**Correctness:** 4
**Technical Novelty And Significance:** 3
**Empirical Novelty And Significance:** 3
**Recommendation:** 5

**Clarity, Quality, Novelty And Reproducibility:**

$\textbf{Clarity}$:
This paper is clearly written. However, the description of the asymptotic equivariance of SO(3)/SO(2) is a little bit abstract, it is better to illustrate such equivariance with figure. Besides, it would be benefit to give the pseudo-code of the implementation of the custom sparse gradients.

$\textbf{Quality}$:
This paper is technically sound, for example, the custom sparse gradients which are not supported by TensorFlow and Pytorch, are implemented by the authors.

$\textbf{Novelty and originality}$:
This paper is original and novel, proposing a new model to overcome the shortcoming that exist in continuous spherical CNNs and discrete spherical CNNs.



**Strength And Weaknesses:**

Two main advantages of this paper are as follows:

1, The paper is well written and the key contribution is easy to understand.

2, The analysis of equivariance error, computational cost and memory usage are provided to support the efficiency of the methods.

Two main disadvantages of this paper are as follows:

1, In experiments, the state-of-the-art claim seems unprecise. For example, in 2D3DS dataset, the HoHoNet (https://arxiv.org/pdf/2011.11498v3.pdf) even achieve mIoU of 52%. Besides, the improvement is minor without data augmentation compared to previous methods.

2, The sufficiency of the restriction from SO(3) to SO(3)/SO(2) equivariance need to be further developed. It seems that the method can’t capture the fully SO(3) equivariance as continuous approaches, which may lead to performance degradation on dataset defined on sphere with random rotation.


**Summary Of The Paper:**

This paper propose the hybrid discrete-continuous group convolution on sphere that possess both equivariant property and computational scalability. The sparse tensor representation is used in implementation to further save computation cost and memory. Experiments are performed on spherical MNIST, Omni-SYNTHIA, 2D3DS and Pano3D, which shows some improvements over previous works.



**Summary Of The Review:**

In summary, this paper is well written, but it still needs to be refined carefully. I recommend marginally below the acceptance threshold for this paper.

---

> ### Author Response · Authors · 2022-11-18
> **Response to Reviewer D1fT (1 of 2)**
>
> We thank the Reviewer for their comments.  We respond to each comment in turn below.  The Reviewer's original comments are italicized, while our responses are given in Roman font.  All revisions to the manuscript are highlighted in red.
>
> *1, In experiments, the state-of-the-art claim seems unprecise. For example, in 2D3DS dataset, the HoHoNet (https://arxiv.org/pdf/2011.11498v3.pdf) even achieve mIoU of 52%.*
>
> Thank you for bringing this paper to our attention.  We have included results from HoHoNet in the results tables for 2D3DS (Table 2).  We noticed that experiments with HoHoNet are performed at various resolutions.  For a fair comparison results should clearly be compared for experiments at the same resolution.  We have therefore appended to these tables the effective resolution of the experiment $\tilde{L}$.  When considering the same resolution, DISCO still achieves SOTA on 2D3DS.
>
> *Besides, the improvement is minor without data augmentation compared to previous methods.*
>
> This comment perhaps relates to experiments at different resolutions.  Even for the setting without augmentation we see an improvement of ~2% in mIoU and ~7% mAcc for 2D3DS when compared to the best alternatives.
>
> *2, The sufficiency of the restriction from SO(3) to SO(3)/SO(2) equivariance need to be further developed. It seems that the method can’t capture the fully SO(3) equivariance as continuous approaches, which may lead to performance degradation on dataset defined on sphere with random rotation.*
>
> To the contrary, full SO(3) equivariance can be captured by the DISCO approach in two ways.
>
> For the first approach, directional filters can be rotated by the usual $SO(3)$ rotations.  In this setting full $SO(3)$ rotational equivariance is achieved.  This is discussed in Section 3.3 under the subheading **$SO(3)$ rotational equivariance**.  However, we do not implement this approach due to the computational cost of $SO(3)$ convolutions.  In this scenarios not only is the cost of $S^2 \rightarrow SO(3)$ convolution increased but convolutions of subsequent layers must be performed on $SO(3)$.  While efficient DISCO convolutions on $SO(3)$ can certainly be considered (in an analogous manner to Cohen et al. 2018 and Cobb et al. 2021) and will satisfy $SO(3)$ rotational equivariance, computational cost will be further increased.  In the current manuscript we focus on highly computationally scalable approaches hence we do not implement this approach or consider it further.
>
> For the second approach to achieve full SO(3) equivariance, axisymmetric filters may be considered.  In this setting, since the filters are invariant to rotations about their own axis, spherical convolutions with $SO(3)$ and $SO(3)/SO(2)$ rotations are equivalent and so full $SO(3)$ rotational equivariance is achieved.  This is discussed in Section 3.3 under the subheading **Axisymmetric filters and $SO(3)$ rotational equivariance**.  Since the axisymmetric spherical convolution is efficient and outputs remain on $S^2$, we do implement this approach in the DISCO framework.  Indeed, we perform numerical experiments that show that excellent $SO(3)$ rotational equivarance is achieved in this setting, with equivariance errors at the level of $\sim 10^{-4}$ for single precision arithmetic (see Section 5.1 and Appendix D [previously Appendix C]).  Furthermore, for the MNIST experiments where full $SO(3)$ rotational equivariance is desired we adopt axisymmetric filters and demonstrate excellent rotational invariance of the complete architecture (see Section 5.2 and Appendix E.1 [previously Appendix D.1]).  Axisymmetric filters are clearly less expressive than directional filters, nevertheless we and others (Esteves et al. 2018) both show axisymmetric filters can be highly effective.
>
> Finally, it one wishes to achieve the expressivity of directional filter kernels, with maximal computational efficiency, then directional filters with $SO(3)/SO(2)$ rotations may be considered, which we show leads to asymptotic $SO(3)/SO(2)$ equivariance.  We stress that this setting is not essential and full $SO(3)$ equivariance can be achieved using either of the approaches discussed above if desired.  Nevertheless, $SO(3)/SO(2)$ equivariance can be desirable in many practical applications since content in spherical signals is often orientated and similar content often appears at similar latitudes, particularly for $360{}^\circ$ panoramic photos and video.  We find in numerical experiments that asymptotic $SO(3)/SO(2)$ equivariance, coupled with the expressivity of directional filter kernels, performs very well in benchmark problems.

---

> > ### Author Response · Authors · 2022-11-18
> > **Response to Reviewer vRza (2 of 2)**
> >
> > *This paper is clearly written. However, the description of the asymptotic equivariance of SO(3)/SO(2) is a little bit abstract, it is better to illustrate such equivariance with figure.*
> >
> > We have added a new diagram (Figure 3) in Appendix D [previously Appendix C] to illustrate $SO(3)$ and $SO(3) / SO(2)$ rotations, which we also refer to in the main text.  Equivariance is simply with respect to these different rotations.
> >
> > *Besides, it would be benefit to give the pseudo-code of the implementation of the custom sparse gradients.*
> >
> > Pseudo-code outlining the custom sparse gradient approach has been added to Appendix C [previously Appendix B].

---

> > > ### Comment · Reviewer_D1fT · 2022-11-21
> > > **Thanks for your elaborate response.**
> > >
> > > Thanks for your elaborate response.  I’m glad to see the supply of the Figure 3 and pseudo code. That make the paper more clear. I agree with Reviewer vRza that the there’s limitation on SO(3)/SO(2) equivariance and the high-resolution experiments should be carried out as comparison to previous works.

---

> > > > ### Author Response · Authors · 2022-12-11
> > > > **Response to Reviewer D1fT (second round)**
> > > >
> > > > We thank the Reviewer for their further comments.  We respond to each comment in turn below.  The Reviewer's original comments are italicized, while our responses are given in Roman font.
> > > >
> > > > *Thanks for your elaborate response. I’m glad to see the supply of the Figure 3 and pseudo code. That make the paper more clear. I agree with Reviewer vRza that the there’s limitation on SO(3)/SO(2) equivariance and the high-resolution experiments should be carried out as comparison to previous works.*
> > > >
> > > > In terms of asymptotic SO(3)/SO(2) equivariance, we have studied this again and now realize that the spherical convolution with SO(3)/SO(2) rotations in fact satisfies a form of asymptotic SO(3) equivariance (rather than asymptotic SO(3)/SO(2) equivariance), which we hadn't appreciated previously.  This result is stronger than asymptotic SO(3)/SO(2) equivariance but it weaker than full SO(3) since it remains asymptotic (as $\beta \rightarrow 0$).  In any case, as commented previously, full SO(3) rotational equivariance can be achieved by adopting axisymmetric filters.
> > > >
> > > > In terms of further high-resolution experiments, while we agree these would be interesting, we unfortunately do not have the computational resources to run these experiments during this further discussion period of the review process.  And in any case, we understand that further revisions to the manuscript should not be made at this stage.

---

### Decision · Program_Chairs · 2023-01-20

**Decision:**

Accept: poster

**Justification For Why Not Higher Score:**

Please see the weakness in part 1. The paper is actually marginally acceptable.

**Justification For Why Not Lower Score:**

The average score, 6.5, of the paper makes the paper ranked the 2nd among the papers I handled. There are two strong supports from the reviewers (two 8's) and the negative score (two 5's) are not bad.

**Metareview: Summary, Strengths And Weaknesses:**

The paper originally got one 8 (accept, good paper), one 6 (marginally above threshold) and two 5s (marginally below threshold). The strength includes clear paper writing, detailed analysis, notable theoretical speedup and memory saving etc. The weakness includes not-so-strong novelty and experiments, limitation on SO(3)/SO(2) equivariance, no validation on the claim of memory-and-computation efficiency, etc. After rebuttal, one reviewer raised from 6 to 8. By the overall scores, the AC recommended acceptance.

**Note From Pc:**

if the above contains the word "oral" or "spotlight" please see: "oral" presentation means -> notable-top-5% and "spotlight" means -> notable-top-25%. As stated in our emails, we are disassociating presentation type from AC recommendations

**Summary Of Ac-Reviewer Meeting:**

N/A